# Reduced Policy Optimization for Continuous Control with Hard Constraints

**Shutong Ding**[1]    **Jingya Wang**[1]    **Yali Du**[2]    **Ye Shi**[1] *

[1]ShanghaiTech University    [2]King's College London

{dingsht, wangjingya, shiye}@shanghaitech.edu.cn
yali.du@kcl.ac.uk

## Abstract

Recent advances in constrained reinforcement learning (RL) have endowed reinforcement learning with certain safety guarantees. However, deploying existing constrained RL algorithms in continuous control tasks with general hard constraints remains challenging, particularly in those situations with non-convex hard constraints. Inspired by the generalized reduced gradient (GRG) algorithm, a classical constrained optimization technique, we propose a reduced policy optimization (RPO) algorithm that combines RL with GRG to address general hard constraints. RPO partitions actions into basic actions and nonbasic actions following the GRG method and output the basic actions via a policy network. Subsequently, RPO calculates the nonbasic actions by solving equations based on equality constraints using the obtained basic actions. The policy network is then updated by implicitly differentiating nonbasic actions with respect to basic actions. Additionally, we introduce an action projection procedure based on the reduced gradient and apply a modified Lagrangian relaxation technique to ensure inequality constraints are satisfied. To the best of our knowledge, RPO is the first attempt that introduces GRG to RL as a way of efficiently handling both equality and inequality hard constraints. It is worth noting that there is currently a lack of RL environments with complex hard constraints, which motivates us to develop three new benchmarks: two robotics manipulation tasks and a smart grid operation control task. With these benchmarks, RPO achieves better performance than previous constrained RL algorithms in terms of both cumulative reward and constraint violation. We believe RPO, along with the new benchmarks, will open up new opportunities for applying RL to real-world problems with complex constraints.

## 1   Introduction

The past few years have witnessed the significant success of reinforcement learning (RL) [41] in various fields such as mastering GO [38], robotic manipulations [20, 24], autonomous driving [34], and smart grid controlling [52, 47, 33], etc. However, it is still challenging to deploy RL algorithms in real-world control tasks, such as operating robots on a specific surface, controlling power generation to fulfill the demand, etc. The principal reason here is that hard constraints must be taken into account in these control problems. Concretely, such constraints come in the form of both equality and inequality constraints and can be nonlinear or even nonconvex, which makes it difficult to handle them in RL. Moreover, unlike soft constraints, hard constraints take explicit form and require strict compliance, which poses additional challenges.

---

*Corresponding author.

37th Conference on Neural Information Processing Systems (NeurIPS 2023).

Existing work on constrained RL can be divided into two categories. The first category involves treating constraints as implicit or soft constraints and using Safe RL algorithms [4, 13, 14, 49, 25, 51, 26, 48]. These algorithms approximate the cumulative costs associated with the constraints and optimize the policy network to balance the trade-off between the cumulative rewards and costs. While Safe RL algorithms have provided certain guarantees on soft constraints, they cannot handle equality constraints since the constraints may not be satisfied due to approximation errors. Moreover, handling multiple constraints using Safe RL algorithms can be computationally expensive, and the existing benchmarks generally only involve simple soft constraints that do not reflect the complexity of real-world applications.

In the second category, the approaches treat the output of the policy network as a set of sub-optimal actions, correcting them to satisfy the constraints by adding an extra constrained optimization procedure. This technique has been explored in several works, such as [16, 29, 10, 33, 32, 24]. Compared to Safe RL algorithms, these algorithms can guarantee the satisfaction of hard constraints but are mostly designed for specific applications and hard constraints of a particular form. For instance, OptLayer [29] employs OptNet [8] into RL to ensure the satisfaction of linear constraints in robotic manipulation. As a result, these approaches to RL with hard constraints are limited in their ability to generalize and lack a generalized formulation.

To address the limitations of existing RL algorithms in handling hard constraints, we propose a constrained off-policy reinforcement learning algorithm called Reduced Policy Optimization (RPO). Our approach is inspired by Generalized Reduced Gradient (GRG), a classical optimization method. RPO partitions actions into basic actions and nonbasic actions following the GRG method and uses a policy network to output the basic actions. The nonbasic actions are then calculated by solving equations based on equality constraints using the obtained basic actions. Lastly, the policy network is updated by the reduced gradient with respect to basic actions, ensuring the satisfaction of the equality constraints. Moreover, we also incorporate a modified Lagrangian relaxation method with an exact penalty term into the loss function of the policy network to improve the initial actions. Our approach provides more confidence when deploying off-policy RL algorithms in real-world applications, as it ensures that the algorithms behave in a feasible and predictable manner. It is worth noting that there is currently a lack of RL environments with complex hard constraints. This motivates us to develop three new benchmarks to validate the performance of our proposed method. To summarize, our main contributions are as follows:

1) **Reduced Policy Optimization.** We present RPO, an innovative approach that introduces the GRG algorithm into off-policy RL algorithms. RPO treats the output of the policy network as a good initial solution and enforces the satisfaction of equality and inequality constraints via solving corresponding equations and applying reduced gradient projections respectively. To the best of our knowledge, this is the first attempt to fuse RL algorithms with the GRG method, providing a novel and effective solution to address the limitations of existing RL algorithms in handling hard constraints.

2) **RL Benchmarks with Hard Constraints.** We develop three benchmarks with hard constraints to validate the performance of our method, involving Safe CartPole, Spring Pendulum and Optimal Power Flow (OPF) with Battery Energy Storage. Comprehensive experiments on these benchmarks demonstrate the superiority of RPO in terms of both cumulative reward and constraint violation. We believe that these benchmarks will be valuable resources for the research community to evaluate and compare the performance of RL algorithm in environments with complex hard constraints.

## 2 Related Works

In this section, we first review the existing methods in the field of constrained RL and divide them into different types according to different constraints. We summarize the differences between our method and previous works in Table 1. Besides, we also introduce the literature on the GRG method as the motivation and background of our method.

**Soft-Constrainted RL.** RL with soft constraints is well-studied and also known as safe reinforcement learning (Safe RL). One of the principal branches in Safe RL methods is based on the Lagrangian relaxation such as [13, 39, 22], where the primal-dual update is used to enforce the satisfaction of constraints. Besides, different penalty terms [25, 35, 42] are designed to maintain the tradeoff between the optimality in reward and safety guarantees. Moreover, Notable classical methods of safe reinforcement learning include CPO [4] based on the local policy search; Lyapunov-based

approaches [14]; PCPO [49], FOCOPS [51], CUP [26] based on two-step optimization, EHCSB [46] based on bi-level optimization formulation, and SMPS [9] based on recovery and shielding. Besides, RL with soft instantaneous constraints was first studied separately in [16]. This approach adds a safety layer to the policy with a pre-trained constraint-violation classifier but can only handle linear constraints. Other approaches include [43] based on the Recovery Policy and [45, 44] based on the Gaussian Process. Very recently, Unrolling Safety Layer [50] was proposed to handle the soft instantaneous constraints in RL. However, these approaches in soft-constrained RL tackle constraints implicitly and cannot ensure strict compliance with the constraints, especially the equality ones. By contrast, our method can handle both hard equality and inequality constraints effectively under the RL framework.

**Hard-Constrained RL.** Compared to RL with soft constraints, RL with hard constraints is rarely studied, and most schemes are designed for some specific application. Pham et al. [29] proposed a plug-in architecture called OptLayer based on OptNet [8], which belongs to a special neural network called optimization layer [5, 40], to avoid infeasible actions in robotics manipulation. In studying resource allocation problems, Bhatia et al. [10] developed further Optlayer techniques to deal with hierarchical linear constraints. Liu et al. [24] investigated robotics manipulation tasks based on RL and used manifold optimization to handle the hard constraints with the inverse dynamic model of robots. Other researchers such as [47, 33] incorporated special optimization techniques into RL to handle power operation tasks in smart grids. However, these methods are designed solely for specific applications or constraints of special types. For example, OptLayer can only handle linear constraints in robotics manipulation. By contrast, RPO is not designed for one specific application and can handle general hard constraints in decision-making problems.

**Generalized Reduced Gradient Method.** GRG [3] is a classical constrained optimization technique and has the powerful capability to handle optimization with nonlinear hard constraints [27]. The basic idea of GRG is closely related to the simplex method in linear programming which divides variables into basic and nonbasic groups and then utilizes the reduced gradient to perform the update on the basic variables and nonbasic variables respectively. In the past decades, the GRG method has been applied to stock exchange [6], optimal control in very-large-scale robotic systems [31], optimal power flow models [17], and many other fields. Additionally, more recently, several works such as [37] also fuse the genetic algorithms with the GRG method. Besides, recent research like DC3 [18] in deep learning is also based on the idea of the GRG method. DC3 was proposed for learning to solve constrained optimization problems and has demonstrated its good capability to obtain near-optimal decisions with the satisfaction of nonlinear constraints.

## 3 Preliminaries

**Markov Decision Process.** A classical Markov decision process (MDP) [41] can be represented as a tuple $(S, A, R, P, \mu)$, where $S$ is the state space, $A$ is the action space, $R : S \times A \times S \to \mathbb{R}$ is the reward function, $P : S \times A \times S \to [0, 1]$ is the transition probability function (where $P(s' \mid s, a)$ is the transition probability from the previous state $s$ to the state $s'$ when the agent took action $a$ in $s$), and $\mu : S \to [0, 1]$ is the distribution of the initial state. A stationary policy $\pi : S \to \mathcal{P}(A)$ is a map from states to probability distributions over actions, and $\pi(a|s)$ denotes the probability of taking action $a$ in state $s$. The set of all stationary policies $\pi$ is denoted by $\Pi$. The goal of RL is to find an optimal $\pi^*$ that maximizes the expectation of the discounted cumulative reward, which is $J_R(\pi) = \mathbb{E}_{\tau \sim \pi} \left[ \sum_{t=0}^{\infty} \gamma^t R(s_t, a_t, s_{t+1}) \right]$. Here $\tau = (s_0, a_0, s_1, a_1 \ldots)$ denotes a trajectory, and $\tau \sim \pi$ is the distribution of trajectories when the policy $\pi$ is employed. Then, The value function of

| Constraint | Method | Multiple | Inequality | Equality | Generality | Model Agnostic |
|---|---|---|---|---|---|---|
| Soft, Cumulative | CPO [4], RCPO [42], PCPO [49] | ✗ | ✓ | ✗ | ✓ | ✓ |
| Soft, Cumulative | Lyapunov [14, 15] | ✗ | ✓ | ✗ | ✓ | ✓ |
| Soft, Cumulative | FOCOPS [51], CUP [48] | ✗ | ✓ | ✗ | ✓ | ✓ |
| Soft, Cumulative | IPO [25], P3O [35] | ✓ | ✓ | ✗ | ✓ | ✓ |
| Soft, Cumulative/Instantaneous | Lagrangian [13, 11, 39], FAC [28] | ✗ | ✓ | ✗ | ✓ | ✓ |
| Soft, Instantaneous | Safety Layer [16] | ✓ | ✓ | ✗ | ✗ (Linear) | ✓ |
| Soft, Instantaneous | Recovery RL [43] | ✗ | ✓ | ✗ | ✓ | ✓ |
| Hard, Instantaneous | OptLayer [29], ReCO-RL [10] | ✓ | ✓ | ✓ | ✗ (Specific Linear) | ✓ |
| Hard, Instantaneous | ATACOM [24] | ✓ | ✓ | ✓ | ✗ (Specific Nonconvex) | ✗ (Robotics) |
| Hard, Instantaneous | CC-SAC [33], Hybrid-DDPG [47] | ✓ | ✓ | ✓ | ✗ (Specific Nonconvex) | ✗ (Power Grid) |
| Hard, Instantaneous | RPO(*) | ✓ | ✓ | ✓ | ✓ | ✓ |

Table 1: Comparison among constrained RL algorithms of different categories

state $s$ is $V_R^\pi(s) = \mathbb{E}_{\tau \sim \pi}\left[\sum_{t=0}^{\infty} \gamma^t R\left(s_t, a_t, s_{t+1}\right) \mid s_0 = s\right]$, the action-value function of state $s$ and action $a$ is $Q_R^\pi(s, a) = \mathbb{E}_{\tau \sim \pi}\left[\sum_{t=0}^{\infty} \gamma^t R\left(s_t, a_t, s_{t+1}\right) \mid s_0 = s, a_0 = a\right]$.

**Generalized Reduced Gradient Method.** GRG considers the following nonlinear optimization problem:

$$\min_{\mathbf{x} \in \mathbb{R}^n} f(\mathbf{x}), \text{ s.t. } \mathbf{h}(\mathbf{x}) = \mathbf{0} \in \mathbb{R}^m, \ \mathbf{a} \le \mathbf{x} \le \mathbf{b} \tag{1}$$

The optimization problem (1) is a general formulation of nonlinear optimization since any nonlinear inequality constraints can always be transformed into equality constraints with inequality box constraints by adding slack variables. GRG first partitions the variable $\mathbf{x}$ into the basic variable $\mathbf{x}^B$ and nonbasic variable $\mathbf{x}^N$. Then the reduced gradient with respect to $\mathbf{x}^B$ is derived as follows [27]:

$$\mathbf{r}^T = \nabla_{\mathbf{x}^B} f(\mathbf{x}^N, \mathbf{x}^B) - \nabla_{\mathbf{x}^N} f(\mathbf{x}^N, \mathbf{x}^B) \left[\nabla_{\mathbf{x}^N} \mathbf{h}(\mathbf{x}^N, \mathbf{x}^B)\right]^{-1} \nabla_{\mathbf{x}^B} \mathbf{h}(\mathbf{x}^N, \mathbf{x}^B), \tag{2}$$

Finally, GRG defines the update step as $\Delta \mathbf{x}^B = -\mathbf{r}$ and $\Delta \mathbf{x}^N = -\left[\nabla_{\mathbf{x}^N} \mathbf{h}\right]^{-1} \nabla_{\mathbf{x}^B} \mathbf{h} \Delta \mathbf{x}^B$ to ensure the equality constraints still hold during the iterations. More details of the GRG method can be referred to in supplementary materials.

# 4 Reduced Policy Optimization

**Problem Formulation.** Although simple explicit constraints in neural networks can be easily handled by some specific activation functions (e.g., the Softmax operator for probability Simplex constraints and the ReLU operators for positive orthant constraints), it is hard to make the output of the policy network satisfy general hard constraints, especially for nonlinear and nonconvex constraints. In this section, we propose RPO to handle MDP with hard constraints formulated as follows:

$$\begin{aligned} \max_{\theta} \quad & J_R(\pi_\theta) \\ \text{subject to} \quad & f_i\left(\pi_\theta(s_t); s_t\right) = 0 \quad \forall i, t, \\ & g_j\left(\pi_\theta(s_t); s_t\right) \le 0 \quad \forall j, t, \end{aligned} \tag{3}$$

where $f_i$ and $g_j$ are the hard deterministic constraints that are only related to $s_t$ and $a_t$ in the current step, and they are required to be satisfied in all states for the policy. Notably, while this formulation is actually a special case of CMDP [7], it focuses more on the hard instantaneous constraints in constrained RL and is different from the cases [4] considered by previous works in Safe RL, where the constraints only involve implicit inequality ones. Besides, it is also possible to transform the action-independent constraints like $g_j(s_{t+1}) \le 0$ into $g_j\left(\pi_\theta(s_t); s_t\right) \le 0$ in some real-world applications if the transition function is deterministic.

RPO consists of a policy network combined with an equation solver to predict the initial actions and a post-plugged GRG update procedure to generate feasible actions under a differentiable framework from end to end. Specifically, the decision process of RPO can be decomposed into a construction stage and a projection stage to deal with equality and inequality constraints respectively. In addition, we also developed practical implementation tricks combined with a modified Lagrangian relaxation method in order to further enforce the satisfaction of hard constraints and fuse the GRG method into RL algorithms appropriately. The pipeline of RPO is shown in Figure 1.

## 4.1 Construction Stage to Handle Equality Constraints

Recalling the nature of equality constraints, it can be viewed as the reduction of the freedom degree in actions. Hence, we follow the formulation in GRG method and divide actions $a \in \mathbb{R}^n$ into basic actions $a^B \in \mathbb{R}^m$ and nonbasic actions $a^N \in \mathbb{R}^{n-m}$, where $n - m$ is defined as the number of equality constraints. Hence, the actual actions that we need to determine are the basic actions. Given that, we utilize the policy network to output this part of actions and then calculate the nonbasic actions, via solving a set of equations defined by equality constraints and the predicted basic actions, to guarantee the satisfaction of equality constraints. Additionally, we also present a correct gradient flow based on GRG method, which makes it possible to train the policy network in an end-to-end way. As shown in Proposition 1, we illustrate how to backpropagate from the nonbasic actions $a^N$ to the basic actions $a^B$.

**Proposition 1.** *(Gradient Flow in Construction Stage) Assume that we have $(n-m)$ equality constraints denoted as $F(a; s) = \mathbf{0}$ in each state $s$. Let $a^B \in \mathbb{R}^m$ and $a \in \mathbb{R}^n$ be the basic actions*

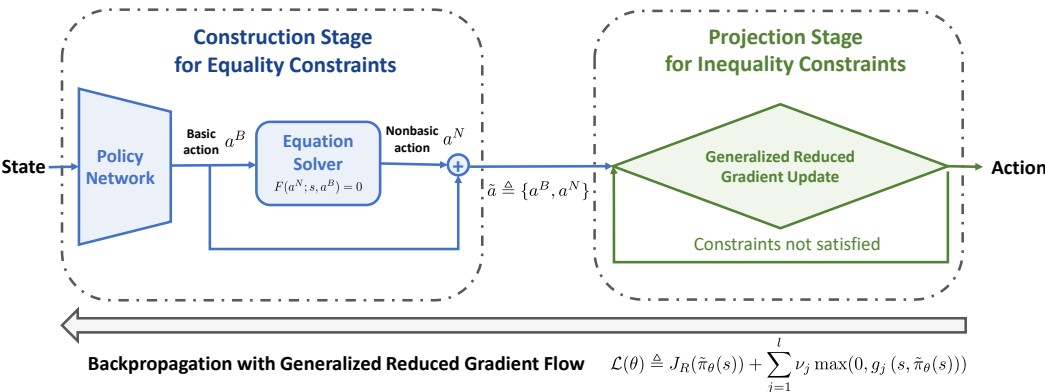

Figure 1: The training and decision procedure of RPO.

and integrated actions, respectively. Let $a^N = \phi_N(a^B) \in \mathbb{R}^{n-m}$ denote the nonbasic actions where $\phi_N$ is the implicit function that determined by the $n - m$ equality constraints. Then, we have

$$\frac{\partial \phi_N(a^B)}{\partial a^B} = -\left(J^F_{:,m+1:n}\right)^{-1} J^F_{:,1:m} \qquad (4)$$

where $J^F = \frac{\partial F(a;s)}{\partial a} \in \mathbb{R}^{(n-m) \times n}$, $J^F_{:,1:m}$ is the first $m$ columns of $J^F$ and $J^F_{:,m+1:n}$ is the last $(n - m)$ columns of $J^F$.

The proof is provided in supplementary materials. Additionally, we denote the decision process of the construction stage, including the policy network $\mu_\theta$ and the equation solving procedure $\phi_N$ as $\tilde{\pi}_\theta$, and initial actions $\tilde{a}$ as the concatenation of $a^B$ and $a^N$, i.e., $\tilde{a} = \left(a^B, a^N\right)$. Furthermore, with respect to the equation-solving procedure, we can always apply the modified Newton's method [27] in the construction stage. However, for some special equality constraints like linear constraints, we can obtain the analytical solutions directly instead of using the modified Newton's method.

## 4.2 Projection Stage to Handle Inequality Constraints

After the construction stage, the initial actions denoted as $\tilde{a}$ may still fall out of a feasible region determined by inequality constraints. Hence, the principal difficulty here is to project the action into the feasible region without violating the equality constraints that have been done in the construction stage. To address this issue, the proposed projection stage is to correct the action in the null space defined by equality constraints. Specifically, this projection is an iterative procedure similar to GRG method, which regards the summation of violation in inequality constraints as the objective $\mathcal{G}(a^B, a^N) \triangleq \sum_j \max\{0, g_j(a; s)\}$. Since the update may destroy the satisfaction of equality constraints, we also need to consider equality constraints in the optimization problem. Then, we can employ GRG updates to the action until all the inequality constraints are satisfied, which means we find the optimal solution to this optimization problem illustrated above. Here the GRG update is defined as

$$\nabla_{a^B} \mathcal{G}(a^B, a^N) \triangleq \frac{\partial \mathcal{G}(a^B, a^N)}{\partial a^B} + \frac{\partial \phi_N(a^B)}{\partial a^B} \frac{\partial \mathcal{G}(a^B, a^N)}{\partial a^N}$$

$$\Delta a^B = \nabla_{a^B} \mathcal{G}(a^B, a^N), \quad \Delta a^N = \frac{\partial \phi_N(a^B)}{\partial a^B} \Delta a^B. \qquad (5)$$

Then, the GRG updates are conducted as follows,

$$a^B_{k+1} = a^B_k - \eta_a \Delta a^B, \quad a^N_{k+1} = a^N_k - \eta_a \Delta a^N. \qquad (6)$$

where $\eta_a$ called projection step can be viewed as the learning rate of GRG update to control the step of update and $a_k$ denotes the actions $a$ at the $k$-th update where $a_0 \triangleq \tilde{a}$. After the projection stage, we obtain the feasible actions that can be deployed in the environment and denote the whole process of the construction stage and projection stage as $\pi_\theta$, which represents the complete policy performed.

**Theorem 1.** *(GRG update in Tangent Space) If $\Delta a^N = \frac{\partial \phi_N(a^B)}{\partial a^B} \Delta a^B$, the GRG update for inequality constraints is in the tangent space of the manifold determined by linear equality constraints.*

The proof is referred to in supplementary materials. Theorem 1 indicates that the projection stage will not violate the linear equality constraints. With regard to nonlinear equality constraints, we can approximate this nonlinear equality manifold with the tangent space at each GRG update. In that case, we need to set the projection step $\eta_a$ with a sufficiently small value, practically, smaller than $10^{-3}$, to avoid destroying the satisfaction of equality constraints. In this way, we can ensure that the GRG update is conducted in the manifold defined by the nonlinear equality constraints. Additionally, a similar projection method was proposed in [18] with $\ell_2$ norm objective, which is likely to fail in the nonlinear situation as we analyze in the supplementary materials.

### 4.3 Practical Implementation

Here we present details about the implementation of our RPO model and some training designs. They mainly include two aspects: 1) how to achieve better initial actions from the policy network and 2) how to incorporate such a decision process into the training of off-policy RL algorithms.

---

**Algorithm 1** Training Procedure of RPO

---

**Input:** Policy network $\mu_\theta(s)$, value network $Q_\omega(s, a)$, penalty factors $\nu$, replay buffer $\mathcal{D}$.

 1: **for** $t$ **in** $1, 2, \cdots, T$ **do**
 2:     Sample the basic actions $a_t^B$ with the output of $\mu_\theta(s_t)$ and some random process.
 3:     Calculate the action $a_t$ according to the construction stage 4.1 and projection stage 4.2.
 4:     Take the action $a_t$ in the environment and store the returned transition in $\mathcal{D}$.
 5:     Sample a mini-batch of transitions in $\mathcal{D}$.
 6:     Update the parameters of the policy network using 8 and the penalty factors using 9.
 7:     Construct TD target as $y_t = r_t + \gamma Q_\omega(s_{t+1}, \pi_\theta(s_{t+1}))$.
 8:     Update the parameters of the value network using MSE loss.
 9: **end for**

---

**Policy Loss with Modified Lagrangian Relaxation.** While the above two-stage decision procedure can cope with the satisfaction of equality and inequality constraints, the policy network is also required to guarantee certain feasibility of inequality constraints for better initial actions $\tilde{a}$. Otherwise, we may need hundreds of GRG updates in the projection stage to satisfy all the inequality constraints. Here, we use an augment loss with Lagrangian relaxation for the policy network to obtain initial actions for the inequality projection stage. Common approaches such as [50] use fixed Lagrangian multipliers in their loss function. However, such fixed Lagrangian multipliers are not easy to obtain and may require extra information and computation to tune. In this paper, we perform a dual update on Lagrangian multipliers to adaptively tune them during the training period. As illustrated in Section 3, after the equality construction stage the constrained MDP problem is formulated as

$$
\begin{aligned}
\max_\pi \quad & J_R(\tilde{\pi}_\theta) \\
\text{subject to} \quad & g_j\left(\tilde{\pi}_\theta(s_t); s_t\right) \leq 0, \ \forall j, t.
\end{aligned}
\tag{7}
$$

This means we need to deal with instantaneous inequality constraints in all states. Hence, we cannot apply the primal-dual update method directly like PDO [13]. Otherwise, we need to compute the dual variables on all states, which is obviously unrealistic. Fortunately, we can maintain only one Lagrangian multiplier $\nu^j$ for each inequality constraint in all states with the exact penalty term $\max\{0, g_j(\tilde{\pi}(s_t); s_t)\}$ [27]. Accordingly, the new objective with the exact penalty term is

$$
\min_\theta \tilde{\mathcal{L}}(\theta) \triangleq -J_R(\tilde{\pi}_\theta) + \mathbb{E}_{s \sim \pi}\left[\sum_j \nu^j \max\{0, g_j(\tilde{\pi}(s_t); s_t)\}\right]
\tag{8}
$$

The following theorem establishes the equivalence between the unconstrained problem (8) and the constrained problem (7).

**Theorem 2.** *(Exact Penalty Theorem) Assume $\nu_s^j$ is the Lagrangian multiplier vector corresponding to $j$th constraints in state $s$. If the penalty factor $\nu^j \geq \|\nu_s^j\|_\infty$, the unconstrained problem (8) is equivalent to the constrained problem (7).*

The proof is referred to in the supplementary materials.

According to the

$$\nu_{k+1}^j = \nu_k^j + \eta_\nu^j \mathbb{E}_{s \sim \pi} \left[ \max \left\{ 0, g_j(\tilde{\pi}(s_t); s_t) \right\} \right] \tag{9}$$

where $\eta_\nu^j$ is the learning rate of $j$-th penalty factors, $\nu_k^j$ is value of $\nu^j$ in the $k$-th step and $\nu_0^j = 0$. Since the exact penalty term, $\mathbb{E}_{s \sim \pi} \left[ \max \left\{ 0, g_j(\tilde{\pi}(s_t); s_t) \right\} \right]$, is always non-negative, the penalty factors are monotonically increasing during the training procedure. Hence, we can always obtain sufficiently large $\nu^j$ that satisfies the condition in Theorem 2, i.e., $\nu^j \geq \|\nu_s^j\|_\infty$. Besides, we also find that the adaptive penalty term does not prevent the exploration for higher rewards of the RL agent at the beginning of the training procedure in Section 5.

**Off-policy RL Training.** Since we augment the policy loss function with the exact penalty term, the actual objective of our algorithm is two-fold. One is to obtain the optimal policy with the satisfaction of hard constraints. Another is to reduce the number of GRG updates performed during the projection stage. This indicates there exists a gap between the behavioral policy and the target policy in RPO, which results from the changing times of GRG updates performed in the projection stage.

Hence, RPO should be trained like off-policy RL methods as Algorithm 1. Specifically, we regard the initial actions $\tilde{a}$ output by the construction stage as the optimization object rather than the actions $a$ post-processed by the projection stage. Otherwise, the training process will be unstable and even collapse due to the changing times of GRG updates in the projection stage, which can also be viewed as a changing network architecture. Besides, we use the $y_t = r_t + \gamma Q_\omega(s_{t+1}, \pi_\theta(s_{t+1}))$ to construct the TD target since $\pi_\theta$ is the actual policy we deploy in the environment.

## 5 Experiments

To validate our method and further facilitate research for MDP with hard constraints, we develop three benchmarks with visualization according to the dynamics in the real world, ranging from classical robotic control to smart grid operation. They involve Safe CartPole, Spring Pendulum, and Optimal Power Flow with Battery Energy Storage. Then, we incorporate RPO into two classical off-policy RL algorithms, DDPG [23] and SAC [21], which we call RPO-DDPG and RPO-SAC respectively.

RPO is compared with three representative Safe RL algorithms, including CPO [4], CUP [48], and Safety Layer [16]. Notably, we transform the hard equality constraints into two inequality constraints since existing Safe RL methods cannot handle both general equality and inequality constraints. Furthermore, we also contrast the RPO-DDPG and RPO-SAC with DDPG-L and SAC-L, where DDPG-L and SAC-L represent DDPG and SAC only modified with the Lagrangian relaxation method we mentioned in Section 4.3 and without the two-stage decision process in RPO respectively. Besides, DDPG-L and SAC-L deal with the equality constraints as we mentioned in Safe RL algorithms. More details related to the concrete RPO-DDPG and RPO-SAC algorithms are illustrated in the supplementary materials. Our code is available at: `https://github.com/wadx2019/rpo`.

### 5.1 RL Benchmarks with Hard Constraints

Specifically, our benchmarks are designed based on [12], with extra interfaces to return the information of the hard constraints. To the best of our knowledge, it is the first evaluation platform in RL that considers both equality constraints and inequality constraints. Figure 2 shows the visualization of these three benchmarks, and the simple descriptions for them are presented in the contexts below. More details about them are provided in the supplementary materials.

**1) Safe CartPole.** Different from that standard CartPole environment in Gym [12], we control two forces from different directions in the Safe CartPole environment. The goal is to keep the pole upright as long as possible while the summation of the two forces should be zero in the vertical direction and be bounded by a box constraint in the horizontal direction. That is, the former is the hard equality constraint, while the latter is the hard inequality constraint.

**2) Spring Pendulum.** Motivated by the Pendulum environment [12], we construct a Spring Pendulum environment that replaces the pendulum with a light spring, which connects the fixed point and the ball. In order to keep the spring pendulum in the upright position, two torques are required to apply in both vertical and horizontal directions. Meanwhile, the spring should be maintained at a fixed length, which introduces a hard equality constraint. Unlike that in Safe CartPole, here equality constraint is

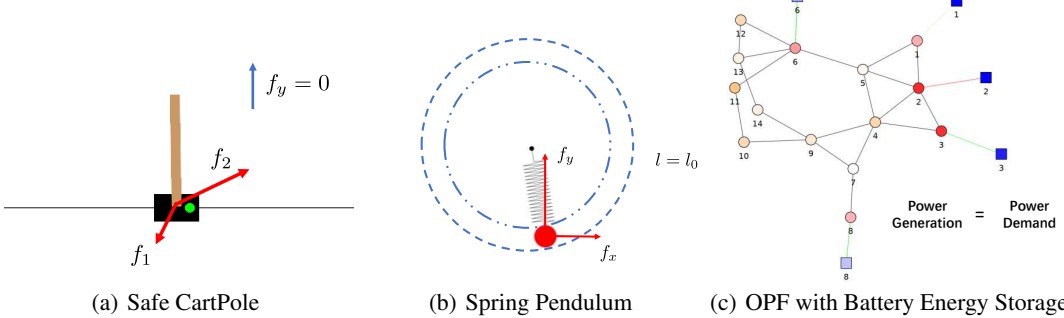

(a) Safe CartPole        (b) Spring Pendulum        (c) OPF with Battery Energy Storage

Figure 2: Visualization of RL benchmarks with hard constraints. (a) **Safe CartPole.** The indicator lamp in the cart will be green if the given actions are feasible, and will be red otherwise. (b) **Spring Pendulum.** The length of the spring will be changed if the equality constraint is violated. (c) **OPF with Battery Energy Storage.** The circle nodes represent the buses in the electricity grid, and the square nodes represent the batteries connected to the generator buses. We use light and shade to reflect the state of batteries and the power generation and demand of buses. In addition, the edge between the generator bus and the battery will be red if the battery is charging, and green if the battery is discharging.

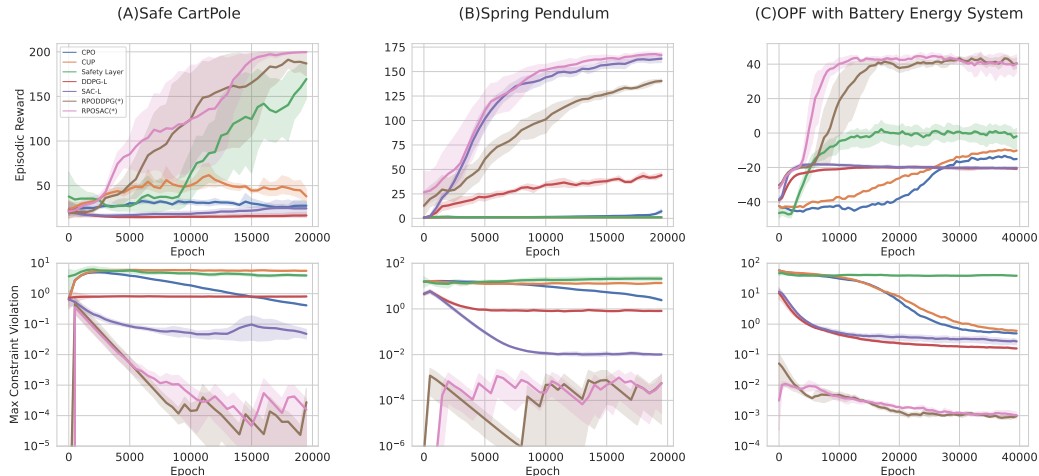

Figure 3: Learning Curves of different algorithms on our three benchmarks across 5 runs. The x-axis is the number of training epochs. The y-axis is the episodic reward (the first line), and max instantaneous constraint violation (the second line) respectively.

state-dependent since the position of the spring is changed during the dynamical process. Besides, the summation of the two torques is also bounded by introducing a quadratic inequality constraint.

**3) Optimal Power Flow with Battery Energy Storage.** Optimal Power Flow (OPF) is a classical problem in smart grid operation to minimize the total cost of power generation with the satisfaction of grid demand. However, battery energy storage systems or electric vehicles have been integrated into smart grids to alleviate the fluctuations of renewable energy generation recently. As illustrated in [30, 36], we need to jointly optimize the charging or discharging of batteries with OPF for the long-term effect of the smart grid operation. According to this real-world application, we design this environment, which is possessed of nonlinear power flow balance equality constraints and inequality box constraints on the actions for feasibility.

## 5.2 Evaluation Metrics

We use three metrics to evaluate the algorithms as follows:

**Episodic Reward:** cumulative reward in a whole episode.

| Task | Metrics | CPO | CUP | Safety Layer | DDPG-L | SAC-L | RPO-DDPG(*) | RPO-SAC(*) |
|---|---|---|---|---|---|---|---|---|
| Safe CartPole | Ep. reward | 20.1000(2.9138) | 63.9000(15.0629) | 53.8000(25.7364) | 15.5000(1.7464) | 40.5000(15.5772) | 200.0000(0.0000) | 200.0000(0.0000) |
| | Max In. ineq | 0.0000(0.0000) | 0.0000(0.0000) | 0.0086(0.0153) | 0.0000(0.0000) | 0.0000(0.0000) | 0.0000(0.0000) | 0.0000(0.0000) |
| | Max In. eq | 0.0389(0.0155) | 0.5344(0.0148) | 1.7099(1.8248) | 0.0408(0.0055) | 0.0487(0.0126) | 0.0000(0.0000) | 0.0000(0.0000) |
| | Max Ep. ineq | 0.0000(0.0000) | 0.0000(0.0000) | 0.7904(1.6355) | 0.0000(0.0000) | 0.0000(0.0000) | 0.0000(0.0000) | 0.0000(0.0000) |
| | Max Ep. eq | 0.1096(0.0217) | 0.0000(0.0000) | 13.6603(0.0000) | 0.1180(0.0131) | 0.1038(0.0645) | 0.0000(0.0000) | 0.0000(0.0000) |
| Spring Pendulum | Ep. reward | 15.4687(12.7866) | 0.8599(0.5242) | 1.1155(0.6959) | 76.4383(27.4531) | 149.6762(4.2608) | 175.1558(15.1087) | 182.4221(2.8414) |
| | Max In. ineq | 0.0000(0.0000) | 0.0000(0.0000) | 6.0975(6.8860) | 0.0000(0.0000) | 0.0000(0.0000) | 0.0000(0.0000) | 0.0000(0.0000) |
| | Max In. eq | 0.7274(0.4259) | 9.5898(2.6630) | 3.9134(2.8721) | 0.3805(0.0441) | 0.0078(0.0013) | 0.0000(0.0000) | 0.0000(0.0000) |
| | Max Ep. ineq | 0.0000(0.0000) | 0.0000(0.0000) | 36.9545(9.1365) | 0.0000(0.0000) | 0.0000(0.0000) | 0.0000(0.0000) | 0.0000(0.0000) |
| | Max Ep. eq | 1.9064(0.7869) | 16.3114(4.9027) | 25.2101(10.1734) | 1.0905(0.5024) | 0.0837(0.0809) | 0.0000(0.0000) | 0.0000(0.0000) |
| OPF with Battery Energy Storage | Ep. reward | -10.3469(1.3825) | -5.8377(0.9521) | -24.1147(8.9916) | -22.0649(0.9199) | -19.4738(2.0778) | 42.9710(17.8438) | 42.0764(10.1048) |
| | Max In. ineq | 0.0000(0.0000) | 0.0000(0.0000) | 0.0000(0.0000) | 0.0000(0.0000) | 0.0000(0.0000) | 0.0002(0.0004) | 0.0003(0.0005) |
| | Max In. eq | 0.4413(0.0215) | 0.4866(0.0136) | 11.4698(10.7497) | 0.1348(0.0136) | 0.1908(0.0203) | 0.0001(0.0000) | 0.0006(0.0000) |
| | Max Ep. ineq | 0.0000(0.0000) | 0.0000(0.0000) | 0.0000(0.0000) | 0.0000(0.0000) | 0.0000(0.0000) | 0.0042(0.0094) | 0.0046(0.0060) |
| | Max Ep. eq | 0.6821(0.0523) | 0.6827(0.0612) | 69.9568(6.0647) | 0.2845(0.0484) | 0.3994(0.1243) | 0.0003(0.0001) | 0.0002(0.0001) |

Table 2: Mean evaluation performance of different algorithms in the three benchmarks. We compare the performance of RPO (RPODDPG, RPOSAC) with the abridged version of our algorithm (DDPG-L, SAC-L) and other Safe RL algorithms (CPO, CUP, Safety Layer) according to episodic reward and max instantaneous & episodic constraint violation for equality and inequality constraints. Each item in the table is averaged across 10 runs with the standard deviations shown in the parentheses.

**Max Instantaneous Constraint Violation:** maximum instantaneous violation of all constraints. It denotes the feasibility of actions in a state because the feasibility of actions relies on the constraint that is the most difficult to be satisfied.

**Max Episodic Constraint Violation:** maximum episodic violation of all constraints. Similar to Mean Constraint Violation, it denotes the feasibility of actions in a whole episode.

### 5.3 Performance of RPO on Reward and Constraints

We plot the learning curves of CPO, CUP, Safety Layer, DDPG-L, SAC-L, RPO-DDPG, and RPO-SAC in Figure 3. Notably, to validate the importance of the two-stage decision procedure, here SAC-L and DDPG-L are actually the versions of RPO-SAC and RPO-DDPG without this procedure. Given the fairness of our experiments, we apply the same shared hyper-parameters for all the algorithms. This empirical result reflects that existing Safe RL algorithms cannot handle the MDP problems with hard constraints, and our approach outperforms other algorithms in terms of both episodic reward and the max constraint violation. Moreover, the learning curves confirm that RPO can also guarantee certain feasibility during the training period.

Besides the learning curves, Table 2 shows the performance of different algorithms after convergence. To present more details on the constraint violations, here the equality and inequality constraint violations are shown separately. Notably, since the $tanh$ and state-dependent $tanh$ activation function are added to limit the output of the neural network for the box constraints, there is no inequality constraint that needs to be satisfied for DDPG-L, SAC-L, and three Safe RL algorithms. That's why these two algorithms achieve zero violation in the inequality constraints in OPF with Battery Energy Storage. However, it is hard for them to satisfy both equality and inequality constraints exactly in OPF with Battery Energy Storage. This is because this environment is very difficult and involves many complex equality and inequality constraints. Practically, a tolerance 1e-3 is introduced here to evaluate the satisfaction of constraints.

## 6 Limitation and Future Work

We acknowledge that there still exist some limitations in RPO. One is that RPO is time-consuming compared to standard neural networks due to the projection stage, where the GRG updates may need to be performed several times. Another is that the equality equation solver required in the construction stage may need to be either chosen or specially designed with domain knowledge. Hence, accelerating RPO and developing more powerful RL algorithms that can handle hard constraints are under consideration in our future works. Besides, while we only validate RPO in RL benchmarks with hard constraints, our method can also be easily extended to cases with both hard constraints and soft constraints as long as a neural network is utilized to fit the mapping between the state-action pair and the cost. Moreover, RPO can be viewed as a supplement to Safe RL algorithms. In scenarios with both hard instantaneous constraints and soft cumulative constraints, we can always use RPO to handle the hard instantaneous constraints and apply some existing Safe RL methods to handle soft cumulative constraints.

# 7 Conclusion

In this paper, we outlined a novel algorithm called RPO to handle general hard constraints under the off-policy reinforcement learning framework. RPO consists of two stages, the construction stage for equality constraints and the projection stage for inequality constraints. Specifically, the construction stage first predicts the basic actions, then calculates the nonbasic action through an equation-solving procedure, and finally concatenates them as the output of the construction stage. The projection stage applies GRG updates to the concatenated actions until all the inequality constraints are satisfied. Furthermore, we also design a special augmented loss function with the exact penalty term and illustrate how to fuse RPO with the off-policy RL training process. Finally, to validate our method and facilitate the research in RL with hard constraints, we have also designed three benchmarks according to the physical nature of the real-world applications, including Safe CartPole, Spring Pendulum, and Optimal Power Flow with Battery Energy Storage. Experimental results in these benchmarks demonstrate the superiority of RPO in terms of both episodic reward and constraint violation.

## Acknowledgement

This work was supported by NSFC (No.62303319), Shanghai Sailing Program (22YF1428800, 21YF1429400), Shanghai Local College Capacity Building Program (23010503100), Shanghai Frontiers Science Center of Human-centered Artificial Intelligence (ShangHAI), MoE Key Laboratory of Intelligent Perception and Human-Machine Collaboration (ShanghaiTech University), and Shanghai Engineering Research Center of Intelligent Vision and Imaging.

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

# A Proofs

## A.1 Proof of Proposition 1

**Proposition 1.** *(Gradient Flow in Construction Stage) Assume that we have $(n - m)$ equality constraints denoted as $F(a; s) = \mathbf{0}$ in each state $s$. Let $a^B \in \mathbb{R}^m$ and $a \in \mathbb{R}^n$ be the basic actions and integrated actions, respectively. Let $a^N = \phi_N(a^B) \in \mathbb{R}^{n-m}$ denote the nonbasic actions where $\phi_N$ is the implicit function that determined by the $n - m$ equality constraints. Then, we have*

$$\frac{\partial \phi_N(a^B)}{\partial a^B} = -\left(J^F_{:,m+1:n}\right)^{-1} J^F_{:,1:m} \tag{10}$$

*where $J^F = \frac{\partial F(a;s)}{\partial a} \in \mathbb{R}^{(m-n) \times n}$, $J^F_{:,1:m}$ is the first $m$ columns of $J^F$ and $J^F_{:,m+1:n}$ is the last $(n - m)$ columns of $J^F$.*

*Proof.* Considering an equality-constrained optimization problem, the reduced gradient [27] is defined as

$$\mathbf{r}^T = \nabla_{\mathbf{x}^B} f(\mathbf{x}^N, \mathbf{x}^B) + \lambda^T \nabla_{\mathbf{x}^B} \mathbf{h}(\mathbf{x}^N, \mathbf{x}^B), \tag{11}$$

where $\lambda^T$ should satisfy $\nabla_{\mathbf{x}^N} f(\mathbf{x}^N, \mathbf{x}^B) + \lambda^T \nabla_{\mathbf{x}^N} \mathbf{h}(\mathbf{x}^N, \mathbf{x}^B) = 0$. $\mathbf{f}, \mathbf{h}$ denote the objective function and the equality constraints, and $x^B, x^N$ are the basic variables and nonbasic variables respectively.

Finally, we will obtain

$$\mathbf{r}^T = \nabla_{\mathbf{x}^B} f(\mathbf{x}^N, \mathbf{x}^B) - \nabla_{\mathbf{x}^N} f(\mathbf{x}^N, \mathbf{x}^B) \left[\nabla_{\mathbf{x}^N} \mathbf{h}(\mathbf{x}^N, \mathbf{x}^B)\right]^{-1} \nabla_{\mathbf{x}^B} \mathbf{h}(\mathbf{x}^N, \mathbf{x}^B). \tag{12}$$

Similarly, the gradient flow from the nonbasic actions to basic actions is

$$\frac{\partial \phi_N(a^B)}{\partial a^B} = -\left(J^F_{:,m:n}\right)^{-1} J^F_{:,1:m}. \tag{13}$$

$\square$

## A.2 Proof of Theorem 1

**Theorem 1.** *(GRG update in Tangent Space) If $\Delta a^N = \frac{\partial \phi_N(a^B)}{\partial a^B} \Delta a^B$, the GRG update for inequality constraints is in the tangent space of the manifold determined by linear equality constraints.*

*Proof.* Firstly, assume the tangent space of the actions in the manifold defined by the equality constraints is $J^F a = d$ where $a = \left[a^B, a^N\right] = \left[a^B, \phi_N(a^B)\right], a^B \in \mathbb{R}^m, \phi_N(a^N) \in \mathbb{R}^{n-m}$.

According to Proposition 1, we have

$$\frac{\partial \phi_N(a^B)}{\partial a^B} = -\left(J^F_{:,m+1:n}\right)^{-1} J^F_{:,1:m}. \tag{14}$$

Then, we define the projection objective as $\mathcal{G}(a^B, a^N) \triangleq \sum_j \max\left\{0, g_j(a; s)\right\}$ and obtain

$$\nabla_{a^B} \mathcal{G}(a^B, a^N) \triangleq \frac{\partial \mathcal{G}(a^B, a^N)}{\partial a^B} + \frac{\partial \phi_N(a^B)}{\partial a^B} \frac{\partial \mathcal{G}(a^B, a^N)}{\partial a^N}$$
$$\Delta a^B = \nabla_{a^B} \mathcal{G}(a^B, a^N), \ \Delta a^N = \frac{\partial \phi_N(a^B)}{\partial a^B} \Delta a^B. \tag{15}$$

Finally, we have

$$
\begin{aligned}
J^F \Delta a &= \left[ J^F_{:,1:m}, J^F_{:,m+1:n} \right] \left[ \begin{array}{c} \Delta a^B \\ \Delta a^N \end{array} \right] \\
&= \left[ J^F_{:,1:m}, J^F_{:,m+1:n} \right] \left[ \begin{array}{c} \Delta a^B \\ \Delta \phi_N(a^B) \end{array} \right] \\
&= J^F_{:,1:m} \Delta a^B + J^F_{:,m+1:n} \Delta \phi_N(a^B) \\
&= J^F_{:,1:m} \Delta a^B + J^F_{:,m+1:n} \frac{\partial \phi_r(a^B)}{\partial a^B} \Delta a^B \\
&= J^F_{:,1:m} \nabla_{a^B} \mathcal{G}(a^B, a^N) + J^F_{:,m+1:n} \frac{\partial \phi_r(a^B)}{\partial a^B} \nabla_{a^B} \mathcal{G}(a^B, a^N) \\
&= J^F_{:,1:m} \left( \frac{\partial \mathcal{G}(a^B, a^N)}{\partial a^B} + \frac{\partial \mathcal{G}(a^B, a^N)}{\partial \phi_N(a^B)} \frac{\partial \phi_N(a^B)}{\partial a^B} \right) \\
&\quad + J^F_{:,m+1:n} \frac{\partial \phi_r(a^B)}{\partial a^B} \left( \frac{\partial \mathcal{G}(a^B, a^N)}{\partial a^B} + \frac{\partial \mathcal{G}(a^B, a^N)}{\partial \phi_N(a^B)} \frac{\partial \phi_N(a^B)}{\partial a^B} \right) \\
&= J^F_{:,1:m} \frac{\partial \mathcal{G}(a^B, a^N)}{\partial a^B} - J^F_{:,1:m} \frac{\partial \mathcal{G}(a^B, a^N)}{\partial \phi_N(a^B)} \left( J^F_{:,m+1:n} \right)^{-1} J^F_{:,1:m} \\
&\quad - J^F_{:,1:m} \frac{\partial \mathcal{G}(a^B, a^N)}{\partial a^B} + J^F_{:,m+1:n} \frac{\partial \mathcal{G}(a^B, a^N)}{\partial \phi_N(a^B)} \left( J^F_{:,m+1:n} \right)^{-1} J^F_{:,1:m} \\
&= 0.
\end{aligned}
\tag{16}
$$

$\square$

## A.3 Proof of Theorem 2

**Theorem 2.** *(**Exact Penalty Theorem**) Assume $\nu_s^j$ is the Lagrangian multiplier vector corresponding to jth constraints in state s. If the penalty factor $\nu^j \geq \|\nu_s^j\|_\infty$, the unconstrained problem (8) is equivalent to the constrained problem (7).*

*Proof.* Since $\nu^j \geq \|\nu_s^j\|_\infty$, we have

$$
\tilde{\mathcal{L}}(\theta) \geq -J_R(\tilde{\pi}_\theta) + \mathbb{E}_{s \sim \pi} \left[ \sum_j \sum_s \nu_s^j \max\{0, g_j(\tilde{\pi}(s); s)\} \right].
\tag{17}
$$

The equality holds when all the inequality constraints are satisfied. Here

$$
\nu_s^j \triangleq \underset{\nu_s^j}{\arg\max} \left\{ \min_\theta -J_R(\tilde{\pi}_\theta) + \mathbb{E}_{s \sim \pi} \left[ \sum_j \sum_s \nu_s^j \max\{0, g_j(\tilde{\pi}(s); s)\} \right] \right\}.
\tag{18}
$$

Therefore, the unconstrained problem $\min_\theta -J_R(\tilde{\pi}_\theta) + \mathbb{E}_{s \sim \pi} \left[ \sum_j \sum_s \nu_s^j \max\{0, g_j(\tilde{\pi}(s); s)\} \right]$ is equivalent to the constrained problem (7). Let $\theta^\star$ be the optimal solution of the unconstrained problem. Then, we have

$$
\begin{aligned}
\tilde{\mathcal{L}}(\theta^\star) &= -J_R(\tilde{\pi}_{\theta^\star}) + \mathbb{E}_{s \sim \pi} \left[ \sum_j \sum_s \nu_s^j \max\{0, g_j(\tilde{\pi}_{\theta^\star}(s); s)\} \right] \\
&\leq -J_R(\tilde{\pi}_\theta) + \mathbb{E}_{s \sim \pi} \left[ \sum_j \sum_s \nu_s^j \max\{0, g_j(\tilde{\pi}(s); s)\} \right] \\
&\leq \tilde{\mathcal{L}}(\theta)
\end{aligned}
\tag{19}
$$

Hence, the unconstrained problem (8) is equivalent to the constrained problem (7). $\square$

## A.4 Limitations of $\ell_2$ Norm Objectives

Projection with $\ell_2$ norm objectives will never come into the feasible region defined by inequality constraints with a sufficiently small projection step.

Assume that the inequality constraint that needs to be satisfied is $g(a; s) \leq 0$. For a given action $a_k$ that does not satisfy the inequality constraint after $k$ updates, i.e., $g(a_k) > 0$. If $a_k$ comes close enough to the feasible region by using the $\ell_2$ norm objective, we apply a linear approximation on $g(a_k) \approx c^T a_k - b$ and then obtain

$$
\begin{aligned}
g(a_{k+1}) &= g(a_k - \Delta a) \\
&= c^T \left( a_k - \eta_a \nabla_a \| c^T a_k - b \|_2^2 \right) - b \\
&= c^T (a_k - 2\eta_a c(c^T a_k - b)) - b \\
&= c^T (a_k - 2\eta_a g(a_k) c) - b \\
&= g(a_k)(1 - 2\eta_a c^T c).
\end{aligned}
\tag{20}
$$

Obviously, to ensure the satisfaction of the inequality constraint, i.e., $g(a_{k+1}) \leq 0$, we must limit the projection step with $\eta_a \geq 1/(2c^T c)$. However, $\eta_a$ is often a sufficiently small number for the satisfaction of nonlinear equality constraints. In this case, the inequality constraints will never be satisfied with a sufficiently small projection step.

## B GRG algorithm

The GRG algorithm is shown as Algorithm 2.

---
**Algorithm 2** Generalized Reduced Gradient Algorithm

---
**Input:** Optimization problem $\min_{\mathbf{x} \in \mathbb{R}^n} f(\mathbf{x})$, s.t. $\mathbf{h}(\mathbf{x}) = \mathbf{0} \in \mathbb{R}^m$, $\mathbf{a} \leq \mathbf{x} \leq \mathbf{b}$.
**Output:** Optimal solution $\mathbf{x}$.
**Assumptions:**

1. Divide $\mathbf{x}$ into $\mathbf{x} = \left( \mathbf{x}^N, \mathbf{x}^B \right)$, where $\mathbf{x}^N \in \mathbb{R}^m, \mathbf{x}^B \in \mathbb{R}^{n-m}$.

2. If $\left( \mathbf{a}^N, \mathbf{a}^B \right)$ and $\left( \mathbf{b}^N, \mathbf{b}^B \right)$ are the corresponding partitions of $\mathbf{a}, \mathbf{b}$, then $\mathbf{a}^N \leq \mathbf{x}^N \leq \mathbf{b}^N$.

3. The $m \times m$ matrix $\nabla_{\mathbf{x}^N} \mathbf{h} \left( \mathbf{x}^N, \mathbf{x}^B \right)$ is nonsingular at $\mathbf{x} = (\mathbf{x}^N, \mathbf{x}^B)$.

1: Define the reduced gradient (with respect to $\mathbf{x}^N$) as

$\mathbf{r}^T = \nabla_{\mathbf{x}^B} f(\mathbf{x}^N, \mathbf{x}^B) + \lambda^T \nabla_{\mathbf{x}^B} \mathbf{h}(\mathbf{x}^N, \mathbf{x}^B)$, where $\nabla_{\mathbf{x}^N} f(\mathbf{x}^N, \mathbf{x}^B) + \lambda^T \nabla_{\mathbf{x}^N} \mathbf{h}(\mathbf{x}^N, \mathbf{x}^B) = 0$.

2: Obtain $\mathbf{r}^T = \nabla_{\mathbf{x}^B} f(\mathbf{x}^N, \mathbf{x}^B) - \nabla_{\mathbf{x}^N} f(\mathbf{x}^N, \mathbf{x}^B) \left[ \nabla_{\mathbf{x}^N} \mathbf{h}(\mathbf{x}^N, \mathbf{x}^B) \right]^{-1} \nabla_{\mathbf{x}^B} \mathbf{h}(\mathbf{x}^N, \mathbf{x}^B)$.

3: Let

$$
\Delta x_i^B = \begin{cases} -r_i & \text{if } r_i < 0, x_i^B < B^B \text{ or } r_i > 0, x_i^B > a^B \\ 0 & \text{otherwise} \end{cases},
$$

$$
\Delta \mathbf{x}^N = - \left[ \nabla_{\mathbf{x}^N} \mathbf{h} \right]^{-1} \nabla_{\mathbf{x}^B} \mathbf{h} \Delta \mathbf{x}^B.
$$

4: Find $\alpha_1, \alpha_2, \alpha_3$ that respectively satisfy

$$
\begin{aligned}
&\max\{\alpha : \mathbf{a}^N \leq \mathbf{x}^N + \alpha \Delta \mathbf{x}^N \leq \mathbf{b}^N\}, \\
&\max\{\alpha : \mathbf{a}^B \leq \mathbf{x}^B + \alpha \Delta \mathbf{x}^B \leq \mathbf{b}^B\}, \\
&\min\{f(\mathbf{x} + \alpha \Delta \mathbf{x}) : 0 \leq \alpha \leq \alpha_1, 0 \leq \alpha \leq \alpha_2\}.
\end{aligned}
$$

5: Update

$$
\begin{aligned}
\mathbf{x}^B &\leftarrow \mathbf{x}^B + \alpha_3 \Delta \mathbf{x}^B \\
\mathbf{x}^N &\leftarrow \mathbf{x}^N + \alpha_3 \Delta \mathbf{x}^N
\end{aligned}
$$

6: Perform (1)-(5) until convergence.

---

## C Algorithm Details

We present the concrete algorithm about how RPODDPG and RPOSAC work in Algorithm 3 and Algorithm 4.

---

**Algorithm 3** RPO-DDPG

---

**Input:** Initial parameters $\theta, \omega$ of policy network $\mu$ and value network $Q$.

1: Initialize the corresponding target network $\theta' \leftarrow \theta, \omega' \leftarrow \omega$.
2: Initialize replay buffer $\mathcal{D}$.
3: Initialize penalty factor for inequality constraints $\nu \leftarrow \mathbf{0}$.
4: **for** each episode **do**
5:     **for** each environment epoch **do**
6:         Select partial action $a_t^B = \mu_\theta(s_t) + \mathcal{N}(0, \epsilon_t)$.
7:         Employ construction stage by solving the equations defined equality constraints

$$\tilde{a}_t = \begin{bmatrix} a_t^B & \phi_N(a_t^B) \end{bmatrix}.$$

8:         Employ projection stage on the concatenated action $\tilde{a}_t$ to obtain feasible action $a_t$
9:         **for** $k = 1, \cdots, K$ **do**

$$\Delta a_{t,k}^B = \nabla_{a^B} \mathcal{G}(a^B, a^N), \quad \Delta a_{t,k}^N = \frac{\partial \phi_N(a^B)}{\partial a^B} \Delta a_{t,k}^B,$$
$$a_{t,k+1}^B = a_{t,k}^B - \eta_a \Delta a_{t,k}^B, \quad a_{t,k+1}^N = a_{t,k}^N - \eta_a \Delta a_{t,k}^N.$$

10:         **end for**
11:         Execute action $a_t$ and observe reward $r_t$ and next state $s_{t+1}$.
12:         Store the transition $< s_t, a_t, r_t, s_{t+1} >$ in $\mathcal{D}$.
13:         Sample a mini-batch of transitions in $\mathcal{D}$.
14:         Update actor and reward critic networks

$$\theta \leftarrow \theta + \eta_\mu \hat{\nabla}_\theta \mathbb{E}_\mathcal{D} \left[ Q_\omega\left(s_t, \tilde{\pi}_\theta(s_t)\right) - \sum_j \nu^j \max\left\{0, g_j(\tilde{\pi}_\theta(s_t); s_t)\right\} \right],$$
$$\omega \leftarrow \omega - \eta_Q \hat{\nabla}_\omega \mathbb{E}_\mathcal{D} \left[ Q_\omega(s_t, a_t) - (r_t + \gamma Q_{\omega'}\left(s_{t+1}, \pi_{\theta'}(s_{t+1})\right)) \right]^2.$$

15:         Perform dual update on $\nu$

$$\nu_{k+1}^j = \nu_k^j + \eta_\nu^j \mathbb{E}_{s \sim \pi} \left[ \max\left\{0, g_j(\tilde{\pi}(s_t); s_t)\right\} \right] \quad \forall j.$$

16:         Soft update target networks:

$$\theta' \leftarrow \tau\theta + (1 - \tau)\theta',$$
$$\omega' \leftarrow \tau\omega + (1 - \tau)\omega'.$$

17:     **end for**
18: **end for**

---

## D Benchmark Details

In this section, we will illustrate the dynamics of our three benchmarks in detail.

### D.1 Safe CartPole

Different from the CartPole environment in Gym [12], two forces from different directions are controlled in the Safe CartPole environment.

**State.** The state space $|S| \in \mathbb{R}^6$ of Safe CartPole includes the position, velocity, and acceleration of the cart; and the angle, angular velocity, and angular acceleration of the pole.

**Algorithm 4** RPO-SAC

**Input:** Initial parameters $\theta, \omega_1, \omega_2$ of policy network $\mu$ and value network $Q_1, Q_2$, Temperature parameter $\alpha$.

1: Initialize the corresponding target network $\omega_1' \leftarrow \omega_1, \omega_2' \leftarrow \omega_2$.
2: Initialize replay buffer $\mathcal{D}$.
3: Initialize penalty factor for inequality constraints $\nu \leftarrow \mathbf{0}$.
4: **for** each episode **do**
5:     **for** each environment epoch **do**
6:         Select partial action $a_t^B \sim \mu_\theta(a_t^B | s_t)$.
7:         Employ construction stage by solving the equations defined equality constraints

$$\tilde{a}_t = \begin{bmatrix} a_t^B & \phi_N(a_t^B) \end{bmatrix}.$$

8:         Employ projection stage on the concatenated action $\tilde{a}_t$ to obtain feasible action $a_t$
9:         **for** $k = 1, \cdots, K$ **do**

$$\Delta a_{t,k}^B = \nabla_{a^B} \mathcal{G}(a^B, a^N), \quad \Delta a_{t,k}^N = \frac{\partial \phi_N(a^B)}{\partial a^B} \Delta a_{t,k}^B,$$
$$a_{t,k+1}^B = a_{t,k}^B - \eta_a \Delta a_{t,k}^B, \quad a_{t,k+1}^N = a_{t,k}^N - \eta_a \Delta a_{t,k}^N.$$

10:         **end for**
11:         Execute action $a_t$ and observe reward $r_t$ and next state $s_{t+1}$.
12:         Store the transition $< s_t, a_t, r_t, s_{t+1} >$ in $\mathcal{D}$.
13:         Sample a mini-batch of transitions in $\mathcal{D}$.
14:         Update actor and reward critic networks

$$\theta \leftarrow \theta + \eta_\mu \hat{\nabla}_\theta \mathbb{E}_\mathcal{D} \left[ -\alpha \log \left( \mu_\theta(a_t|s_t) \right) + Q_\omega \left( s_t, \tilde{\pi}_\theta(s_t) \right) - \sum_j \nu^j \max \left\{ 0, g_j(\tilde{\pi}_\theta(s_t); s_t) \right\} \right],$$
$$\omega_i \leftarrow \omega_i - \eta_Q \hat{\nabla}_{\theta Q} \mathbb{E}_\mathcal{D} \left[ Q_{\omega_i}(s_t, a_t) - \left( r_t + \gamma Q_{\omega_i'} \left( s_{t+1}, \pi_\theta(s_{t+1}) \right) - \alpha \log \left( \mu_\theta(a_{t+1}|s_{t+1}) \right) \right) \right]^2.$$

15:         Perform dual update on $\nu$

$$\nu_{k+1}^j = \nu_k^j + \eta_\nu^j \mathbb{E}_{s \sim \pi} \left[ \max \left\{ 0, g_j(\tilde{\pi}(s_t); s_t) \right\} \right] \quad \forall j.$$

16:         Soft update target networks:

$$\omega_i' \leftarrow \tau \omega_i + (1 - \tau) \omega_i'.$$

17:     **end for**
18: **end for**

---

**Action.** The action space $|A| \in \mathbb{R}^2$ of Safe CartPole is the sign and magnitude of two forces $f_1, f_2$ from different directions. Specifically, one force is inclined 30 degrees below the $x$ axis, and the other is inclined 60 degrees above the $x$ axis.

**Reward.** The goal of Safe CartPole is similar to the CartPole in Gym [12], which requires the pole to keep upright as long as possible. Therefore, we employ the same reward policy that returns 1 if the pole keeps upright. Otherwise, this episode will end since the pole falls.

**Equality Constraints.** The equality constraint of Safe CartPole is that the summation of two forces in $y$ axis should be zero. That is, we desire to avoid extra friction on the cart, i.e.,

$$f_y := f_1 \sin \theta_1 + f_2 \sin \theta_2 = 0. \tag{21}$$

**Inequality Constraints.** The inequality constraint is that the summation of two forces in the $x$ axis should be bounded by a box constraint, which indicates the physical limitation of the magnitude of the summation force in the $x$ axis.

$$\underline{f}_x \leq f_x := f_1 \cos \theta_1 + f_2 \cos \theta_2 \leq \overline{f}_x. \tag{22}$$

Furthermore, according to [19], we derive the dynamics of Safe CartPole when there exists a force in the vertical direction as shown in (23). Notably, we can always derive the position or angle, velocity or angular velocity in the next step, using the semi-implicit Euler method.

$$N_c = f_y + (m_c + m_p) g - m_p l \left( \ddot{\theta} \sin \theta + \dot{\theta}^2 \cos \theta \right),$$

$$\ddot{\theta} = \frac{g \sin \theta - \frac{\mu_p \dot{\theta}}{m_p l}}{l \left\{ \frac{4}{3} - \frac{m_p \cos \theta}{m_c + m_p} \left[ \cos \theta - \mu_c \operatorname{sgn} (N_c \dot{x}) \right] \right\}} +$$

$$\frac{\cos \theta \left\{ \frac{-f_x - m_p l \dot{\theta}^2 [\sin \theta + \mu_c \operatorname{sgn}(N_c \dot{x}) \cos \theta]}{m_c + m_p} + \mu_c g \operatorname{sgn} (N_c \dot{x}) \right\}}{l \left\{ \frac{4}{3} - \frac{m_p \cos \theta}{m_c + m_p} \left[ \cos \theta - \mu_c \operatorname{sgn} (N_c \dot{x}) \right] \right\}}, \tag{23}$$

$$\ddot{x} = \frac{f_x + m_p l \left( \dot{\theta}^2 \sin \theta - \ddot{\theta} \cos \theta \right) - \mu_c N_c \operatorname{sgn} (N_c \dot{x})}{m_c + m_p},$$

where $N_c$ is the pressure on the cart, $m_c, m_p$ are the mass of the cart and pole, $l$ is the length of the pole, $x, \theta$ are the position of the cart and the angle of the pole respectively, and $\mu_c$ is the dynamic friction coefficient of the cart.

### D.2 Spring Pendulum

Motivated by the Pendulum environment [12], Spring Pendulum environment replaces the pendulum with a light spring, which connects the fixed point and the ball in the end of the spring.

**State.** The state space $|S| \in \mathbb{R}^5$ of Spring Pendulum contains the cosine and sine of the angle, angular velocity, length of the spring, and the change rate in length.

**Action.** The action space $|A| \in \mathbb{R}^2$ of Spring Pendulum is the sign and magnitude of two forces $f_x, f_y$ in the $x, y$ axes. Notably, since the spring pendulum is rotating, the angle between the $x$ or $y$ axis and the spring is changing as well. Thus, this environment is more difficult than Safe CartPole in some sense.

**Reward.** The goal of Spring Pendulum is to keep the spring pendulum in an upright position. The episode will never be done until the maximum time step. Specifically, the reward function is $\frac{1}{1+100|\theta|}$, where $\theta$ is the angle between the spring pendulum and the $y$ axis. That means the agent will achieve a reward of nearly 1 when the spring pendulum is close enough to the upright position. Otherwise, a reward of almost 0 will be returned to the agent.

**Equality Constraints.** The equality constraint of the Spring Pendulum is to limit the change rate of length to zero since the spring pendulum is expected to perform like a normal pendulum without changing the length of the pendulum.

$$\dot{l}_t = \dot{l}_{t-1} + \ddot{l}_t dt = 0. \tag{24}$$

Notably, when $\dot{l}_{t-1} = 0$, the equality constraint will be $\ddot{l} = 0$.

**Inequality Constraints.** The inequality constraint is the magnitude constraint on the summation of these two forces.

$$f_x^2 + f_y^2 \leq \bar{f}^2. \tag{25}$$

Furthermore, to connect each component mentioned above, we derive the dynamics of the Spring Pendulum. Applying Euler-Lagrange Equation $\mathcal{L} = T - V$ to spring pendulum, then we obtain

$$\begin{aligned} V =& mgy + \frac{1}{2} k (l - l_0)^2 \\ =& mgl \cos \theta + \frac{1}{2} k (l - l_0)^2, \\ T =& \frac{1}{2} m v^2 = \frac{1}{2} m \left( \dot{x}^2 + \dot{y}^2 \right) \\ =& \frac{1}{2} m \left( \dot{l}^2 + l^2 \dot{\theta}^2 \right). \end{aligned} \tag{26}$$

Applying the Euler-Lagrange equations for $\theta$ and $l$, we have

$$
\begin{aligned}
\frac{d}{dt}\frac{\partial \mathcal{L}}{\partial \dot{\theta}} &= 2ml\dot{\theta} + ml\ddot{\theta}, \\
\frac{\partial \mathcal{L}}{\partial \theta} &= -mg\sin\theta, \\
\frac{d}{dt}\frac{\partial \mathcal{L}}{\partial \dot{l}} &= m\ddot{l} - ml\dot{\theta}^2 + k(l - l_0), \\
\frac{\partial \mathcal{L}}{\partial l} &= -mg\cos\theta.
\end{aligned}
\tag{27}
$$

Finally, we will obtain the dynamics of the spring pendulum

$$
\begin{aligned}
\ddot{\theta} &= \frac{f_r - 2ml\dot{\theta} - mg\sin\theta}{ml}, \\
\ddot{l} &= \frac{f_s + ml\dot{\theta}^2 - k(l - l_0) - mg\cos\theta}{m},
\end{aligned}
\tag{28}
$$

where $f_r = -f_y \sin\theta + f_x \cos\theta$ and $f_s = f_y \cos\theta + f_x \sin\theta$ are the force perpendicular to and along the spring pendulum.

### D.3 Optimal Power Flow with Battery Energy Storage

In smart grid operation controlling, Optimal Power Flow (OPF) is defined as

$$
\begin{aligned}
\min_{p_g, q_g, v} \quad & p_g^T A p_g + b^T p_g \\
\text{subject to} \quad & \underline{p}_g \leq p_g \leq \overline{p}_g, \\
& \underline{q}_g \leq q_g \leq \overline{q}_g, \\
& \underline{v} \leq |v| \leq \overline{v}, \\
& (p_g - p_d) + (q_g - q_d)\,i = \mathrm{diag}(v)Yv,
\end{aligned}
\tag{29}
$$

where $p_g, q_g \in \mathbb{R}^n$ are the active and reactive power generation of the buses, and $v \in \mathbb{C}^n$ are the voltage of the buses in the grid. $Y \in \mathbb{C}^{n \times n}$ denotes the admittance matrix. $p_d, q_d \in \mathbb{R}^n$ are active and reactive power demand of all buses. Notably, some buses in the electricity grid are not generator buses, and $p_g, q_g$ of these buses will be zero and one of the generator buses is the reference (slack) bus which has a fixed voltage argument $\angle v$. Therefore, the actual dimension of $p_g, q_g$ to determine is the number of generator buses $n^g$. Based on the OPF problem, OPF with Battery Energy Storage is defined as

$$
\begin{aligned}
\min_{p_g, q_g, v, p_b} \quad & \sum_{t=0}^{T} p_g^T(t) A p_g(t) + b^T p_g(t) + c^T(t) p_b(t) \\
\text{subject to} \quad & (p_g(t) - p_d(t) - p_b(t)) + (q_g(t) - q_d(t))\,i = \mathrm{diag}(v(t))Yv(t), \\
& \underline{p}_g \leq p_g(t) \leq \overline{p}_g, \\
& \underline{q}_g \leq q_g(t) \leq \overline{q}_g, \\
& \underline{v} \leq |v(t)| \leq \overline{v}, \\
& \underline{p}_b(t) \leq p_b(t) \leq \overline{p}_b(t).
\end{aligned}
\tag{30}
$$

The additional variables $p_b \in \mathbb{R}^n$ is the charging (positive) or discharging (negative) power of the battery groups, and $c(t)$ represents the cost or the income in the time step $t$. Exactly, we only connect the batteries with the generator buses. Therefore, the actual number of $p_b$ to determine is $n^g$ in this benchmark as well.

Specifically, this benchmark is based on a 14-node power system, which is available in the PYPOWER package. We adopt the same topology of the electricity grid with 5 generator nodes in this benchmark. The data on power demand and day-ahead electricity prices are from [1, 2]. We refer to the distribution of the real-world data in one day, which is shown in Figure 4, and normalize

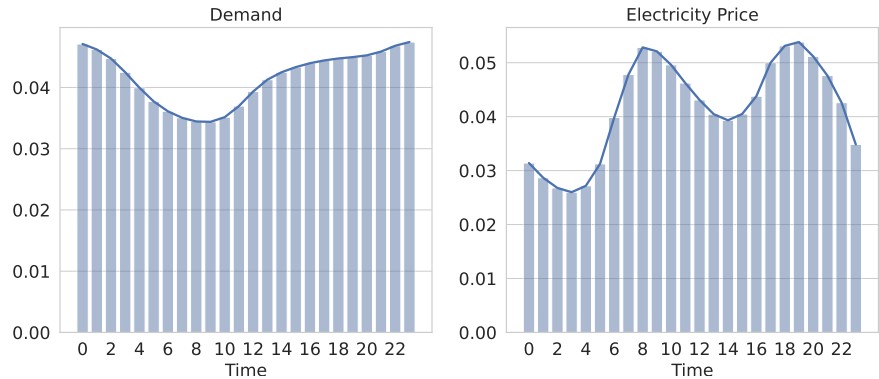

Figure 4: Distribution of power demand (left) and electricity price (right) in 24 hours.

the magnitude of the concrete data to incorporate them into the 14-node power system.

**State.** The state space $|S| \in \mathbb{R}^{57}$ of OPF with Battery Energy Storage contains active and reactive power demand of 14 buses, the state of 5 batteries connected to 5 generator nodes, and the 24-hour day-ahead electricity price.

**Action.** The action space $|A| \in \mathbb{R}^{43}$ of OPF with Battery Energy Storage involves the active and reactive power generation of 5 generator buses, voltage magnitude and argument of 14 buses, and the power generation or demand of 5 batteries. Notably, since the voltage argument in the slack bus is fixed, the actual action space is $|\tilde{A}| \in \mathbb{R}^{42}$. However, for convenience, we include it in the action space and set the fixed value for it in the practical implementation.

**Reward.** The goal of OPF with Battery Energy Storage is to minimize the total cost. Therefore, we regard the negative cost in the current time step $-p_g^T(t)Ap_g(t) - b^Tp_g(t) + c^T(t)p_b(t)$ as the reward. Actually, there exist efficiency parameters $\eta_{ch}, \eta_{dis}$ in the procedure of charging and discharging. Moreover, each part of the cost may have a different magnitude, which needs us to consider a tradeoff between different parts. For convenience, we model all of these factors into $c(t)$.

**Equality Constraints.** The 28 equality constraints of OPF with Battery Energy Storage are the equations of power flow, i.e.,

$$(p_g(t) - p_d(t) - p_b(t)) + (q_g(t) - q_d(t))\, i = \mathrm{diag}(v(t))Y\,v(t). \tag{31}$$

**Inequality Constraints.** The inequality constraint is 58 box constraints on the decision variables.

$$\begin{aligned}
\underline{p}_g &\leq p_g(t) \leq \overline{p}_g, \\
\underline{q}_g &\leq q_g(t) \leq \overline{q}_g, \\
\underline{v} &\leq |v(t)| \leq \overline{v}, \\
\underline{p}_b(t) &\leq p_b(t) \leq \overline{p}_b(t).
\end{aligned} \tag{32}$$

The dynamics of OPF with Battery Energy Storage are much simpler than the former two benchmarks, since the active and reactive power demand $p_d, q_d$ in each time step, is irrelevant to the last state and action. Thus, we only need to care about the change in the electrical power of batteries, i.e., $soc(t) = soc(t-1) + [\eta_{ch}p_{ch}(t) + p_{dis}(t)/\eta_{dis}]$, where $p_{ch}(t) = \max\{0, p_b(t)\}, p_{dis}(t) = \min\{0, p_b(t)\}$.

**Implementation of Construction Stage.** Unlike previous robotic manipulation cases, OPF with Battery Energy Storage environment has nonlinear equality constraints. This means the analytical solution of the corresponding equations cannot be obtained. In addition, while Newton's methods can be applied directly here, it is sometimes unstable empirically. Hence, we followed the idea in

[18], which divides the equation-solving procedure into two steps. The first step is to compute the amplitude of voltage $|v|$ in buses without the generator and voltage argument $\angle v$ in all buses via Newton's methods. Then, we can directly compute the solution of reactive power generation $q_g$ in all buses and $p_g$ in the slack bus in the second step. Moreover, the gradient flow from nonbasic actions to power generation or demand of battery groups $p_b$ can be calculated similarly to the gradient flow from chosen nonbasic actions to active power demand $p_d$.

### D.4 More Explanation on Box Inequality Constraints

Notably, while our three benchmarks only contain box inequality constraints, they are sufficient to evaluate the performance of our RPO algorithm in scenarios with nonlinear inequality constraints. As mentioned in Section 3, any nonlinear inequality constraints can always be transformed into equality constraints plus inequality box constraints by adding slack variables. Hence, evaluating RPO with box inequality constraints and nonlinear equality constraints is sufficient to show RPO's generality. For example, if we have a complex inequality constraint $g(s, a) \leq 0$. Then, we can always transform it into

$$g(s, a) + a^{\text{aug}} = 0, \quad a^{\text{aug}} \leq 0.$$

where $a^{\text{aug}}$ is the slack variable, which can be viewed as the augmented actions in RL.

## E    Additional Experiments

To further discuss the efficiency of RPO, we do some extra experiments in our hardest benchmark, OPF with Battery Energy Storage. The results and related analysis are presented below.

### E.1    Sensitivity Analysis

The projection stage performs the GRG updates until all inequality constraints are satisfied. It is valuable to investigate the impact of the number of maximum GRG updates $K$ on model performance. Here, we conduct experiments under different $K$ values in OPF with the Battery Energy Storage task. As shown in Figure 5, we compare the performance of RPOSAC under $K = 0, 10, 50, 200$ in the projection stage. The result indicates that the

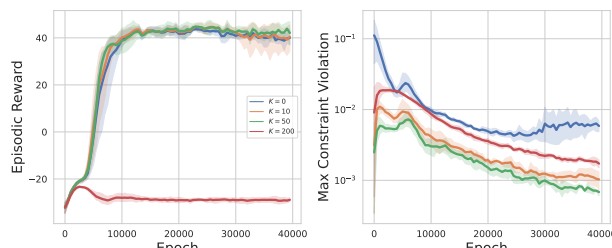

Figure 5: Comparison on RPO under different $K$.

choice of $K$ does not have an obvious impact on the episodic reward when $K$ is not too large. The slight improvement in constraint violation between $K = 10$ and $K = 50$ illustrates that $K = 10$ is actually the most appropriate since it needs much less computation compared with $K = 50$ during the projection stage. Besides, the case of $K = 200$ shows that too large modifications on actions during the training process will lead to poor performance in some situations. The principal reason we believe is that the value network cannot back-propagate an accurate gradient to the policy network when $K$ is too large. Concretely, a large $K$ leads to the samples being too far from the policy $\tilde{\pi}_\theta$, which further results in the inaccurate estimation of the policy $\tilde{\pi}_\theta$ in the value network. Notably, a practical trick is to set a small $K$ during the training period and a large $K$ during the evaluation period.

## E.2 Adaptive v.s. Fixed Penalty Factor

Besides, We conduct experiments to confirm the performance of the adaptive penalty factor with dual update compared to that of the fixed penalty factor as introduced in Section 4. For fairness, we check the converged values of the adaptive penalty factor with learning rate $\eta = 0.02$ in RPOSAC, and we find that the converged penalty factors are ranging around 100. Therefore, we chose $\nu^j = 100$ for all penalty terms in the setting of RPOSAC

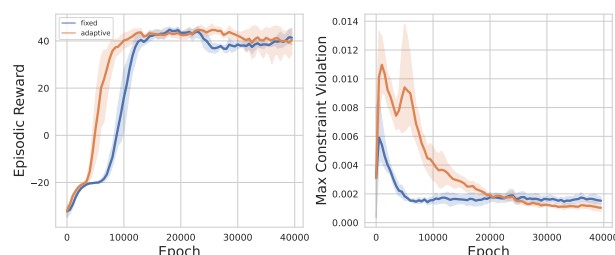

Figure 6: Comparison between RPO with fixed penalty factor and adaptive penalty factor

with the fixed penalty factor. Results in Figure 6 show the advantage of the adaptive penalty factors in terms of episodic reward.

## E.3 Necessity of Backpropagation with Generalized Reduced Gradient

As we illustrate in the construction stage, the integrated actions are actually determined once the basic actions are given. Therefore, it is possible for the value network to approximate the mapping from the basic actions to the nonbasic actions. This seems that there is no need for the backpropagation with generalized reduced gradient which requires extra gradient flow from nonbasic actions if only the hard equality constraints need to be addressed or the hard inequality constraints are only related to the basic actions.

To explore the necessity of the back-propagation with generalized reduced gradient, we also contrast the performance in RPOSAC trained with/without the complete gradient. Concretely, for RPOSAC trained without the complete gradient, we only input the basic actions into the value networks and expect it can approximate the complete gradient, which is known as generalized reduced gradient. The results are shown in Figure 7. It reflects that

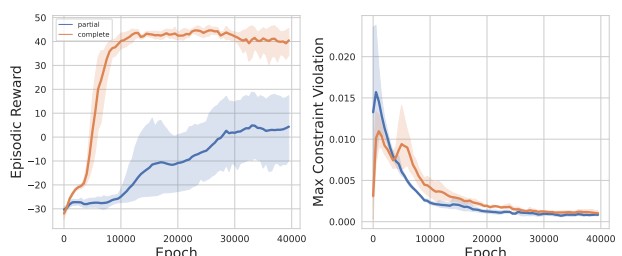

Figure 7: Comparison between RPO with partial gradient and complete gradient.

the value network cannot backpropagate an accurate gradient directly, and indicates the necessity to explicitly construct the gradient flow from nonbasic actions to basic actions.

## E.4 Comparison in Training Time

The training time comparison in our three benchmarks is shown in Tables 3. It can be observed that the training time of RPO-DDPG and RPO-SAC grows linearly with the number of constraints to deal with. However, RPO-DDPG and RPO-SAC still have significant advantages on some of Safe RL algorithms like Safety Layer.

| Task | CPO | CUP | Safety Layer | DDPG-L | SAC-L | RPO-DDPG(*) | RPO-SAC(*) |
|------|-----|-----|--------------|--------|-------|-------------|------------|
| Safe CartPole | 68.7 | 75.2 | 1067.0 | 150.2 | 268.5 | 190.2 | 375.5 |
| Spring Pendulum | 98.6 | 81.4 | 3859.7 | 174.9 | 309.5 | 242.6 | 419.0 |
| OPF with Battery Energy Storage | 288.8 | 239.4 | 6395.0 | 498.9 | 892.7 | 4462.1 | 4828.4 |

Table 3: Training time(s) of different algorithms in the three benchmarks. Each item in the table is averaged across 5 runs.

## F  Hyper-parameters

We implemented our experiments on a GPU of NVIDIA GeForce RTX 3090 with 24GB. Each experiment on Safe CartPole and Spring Pendulum takes about 0.5 hours. Each experiment on OPF with Battery Energy Storage takes about 4 hours. Moreover, we adopt similar neural network architectures for policy and value networks except for the input and output dimensions in all experiments. The policy and value networks both have two hidden layers with 256 hidden units and only differ in input and output layers. Besides, we also show the detailed hyper-parameters used in our experiments. Table 4, Table 5 and Table 6 respectively present the parameters used in Safe CartPole, Spring Pendulum, and OPF with Battery Energy Storage. Additionally, the implementation of three safe RL algorithms in our experiments are based on omnisafe [2] and safe-explorer [3], and recommended values are adopted for hyper-parameters not mentioned in the following tables.

| Parameter | CPO | CUP | Safety Layer | DDPG-L | SAC-L | RPODDPG | RPOSAC |
|---|---|---|---|---|---|---|---|
| Batch Size $\mathcal{B}$ | 256 | 256 | 256 | 256 | 256 | 256 | 256 |
| Discount Factor $\gamma$ | 0.95 | 0.95 | 0.95 | 0.95 | 0.95 | 0.95 | 0.95 |
| Target Smoothing Coefficient $\tau$ | N/A | N/A | 0.005 | 0.005 | 0.005 | 0.005 | 0.005 |
| Frequency of Policy Update | N/A | N/A | N/A | 0.25 | 0.25 | 0.25 | 0.25 |
| Total Epochs $T$ | 2E4 | 2E4 | 2E4 | 2E4 | 2E4 | 2E4 | 2E4 |
| Capacity of Replay Buffer $\mathcal{D}$ | N/A | N/A | 2E4 | 2E4 | 2E4 | 2E4 | 2E4 |
| Random Noise in Exploration $\epsilon$ | N/A | N/A | 1E-2 | 1.0 | N/A | 1.0 | N/A |
| LR for Policy Network $\mu$ | 1E-4 | 1E-4 | 1E-4 | 1E-4 | 1E-4 | 1E-4 | 1E-4 |
| LR for Value Network $Q$ | 3E-4 | 3E-4 | 3E-4 | 3E-4 | 3E-4 | 3E-4 | 3E-4 |
| LR for Penalty Factor $\nu$ | N/A | N/A | N/A | 0.2 | 0.2 | 0.2 | 0.2 |
| Temperature $\alpha$ | N/A | N/A | N/A | N/A | 0.1 | N/A | 0.1 |
| Projection Step $\eta_a$ | N/A | N/A | N/A | N/A | N/A | 2E-2 | 2E-2 |
| Max GRG Updates $K$ | N/A | N/A | N/A | N/A | N/A | 10 | 10 |
| Projection Step in Evaluation $\eta_a^e$ | N/A | N/A | N/A | N/A | N/A | 2E-2 | 2E-2 |
| Max GRG Updates in Evaluation $K^e$ | N/A | N/A | N/A | N/A | N/A | 50 | 50 |

Table 4: Hyper-parameters for experiments in Safe CartPole.

| Parameter | CPO | CUP | Safety Layer | DDPG-L | SAC-L | RPODDPG | RPOSAC |
|---|---|---|---|---|---|---|---|
| Batch Size $\mathcal{B}$ | 256 | 256 | 256 | 256 | 256 | 256 | 256 |
| Discount Factor $\gamma$ | 0.95 | 0.95 | 0.95 | 0.95 | 0.95 | 0.95 | 0.95 |
| Target Smoothing Coefficient $\tau$ | N/A | N/A | 0.005 | 0.005 | 0.005 | 0.005 | 0.005 |
| Frequency of Policy Update | N/A | N/A | N/A | 0.25 | 0.25 | 0.25 | 0.25 |
| Total Epochs $T$ | 2E4 | 2E4 | 2E4 | 2E4 | 2E4 | 2E4 | 2E4 |
| Capacity of Replay Buffer $\mathcal{D}$ | N/A | N/A | 2E4 | 2E4 | 2E4 | 2E4 | 2E4 |
| Random Noise in Exploration $\epsilon$ | N/A | N/A | 1E-2 | 0.5 | N/A | 0.5 | N/A |
| LR for Policy Network $\mu$ | 1E-4 | 1E-4 | 1E-4 | 1E-4 | 1E-4 | 1E-4 | 1E-4 |
| LR for Value Network $Q$ | 3E-4 | 3E-4 | 3E-4 | 3E-4 | 3E-4 | 3E-4 | 3E-4 |
| LR for Penalty Factor $\nu$ | N/A | N/A | N/A | 0.01 | 0.01 | 0.01 | 0.01 |
| Temperature $\alpha$ | N/A | N/A | N/A | N/A | 0.01 | N/A | 0.01 |
| Projection Step $\eta_a$ | N/A | N/A | N/A | N/A | N/A | 2E-3 | 2E-3 |
| Max GRG Updates $K$ | N/A | N/A | N/A | N/A | N/A | 10 | 10 |
| Projection Step in Evaluation $\eta_a^e$ | N/A | N/A | N/A | N/A | N/A | 2E-3 | 2E-3 |
| Max GRG Updates in Evaluation $K^e$ | N/A | N/A | N/A | N/A | N/A | 50 | 50 |

Table 5: Hyper-parameters for experiments in Spring Pendulum.

## G  Systematical Action Division Method

While it is usually not difficult to divide actions into basic and nonbasic actions, we present the systematical action division method below to further show the generality of our method. Firstly, we

---

[2] https://github.com/PKU-Alignment/omnisafe
[3] https://github.com/AgrawalAmey/safe-explorer

| Parameter | CPO | CUP | Safety Layer | DDPG-L | SAC-L | RPODDPG | RPOSAC |
|---|---|---|---|---|---|---|---|
| Batch Size $\mathcal{B}$ | 256 | 256 | 256 | 256 | 256 | 256 | 256 |
| Discount Factor $\gamma$ | 0.95 | 0.95 | 0.95 | 0.95 | 0.95 | 0.95 | 0.95 |
| Target Smoothing Coefficient $\tau$ | N/A | N/A | 0.005 | 0.005 | 0.005 | 0.005 | 0.005 |
| Frequency of Policy Update | N/A | N/A | N/A | 0.25 | 0.25 | 0.25 | 0.25 |
| Total Epochs $T$ | 4E4 | 4E4 | 4E4 | 4E4 | 4E4 | 4E4 | 4E4 |
| Capacity of Replay Buffer $\mathcal{D}$ | N/A | N/A | 2E4 | 2E4 | 2E4 | 2E4 | 2E4 |
| Random Noise in Exploration $\epsilon$ | N/A | N/A | 1E-2 | 1E-4 | N/A | 1E-4 | N/A |
| LR for Policy Network $\mu$ | 1E-4 | 1E-4 | 1E-4 | 1E-4 | 1E-4 | 1E-4 | 1E-4 |
| LR for Value Network $Q$ | 3E-4 | 3E-4 | 3E-4 | 3E-4 | 3E-4 | 3E-4 | 3E-4 |
| LR for Penalty Factor $\nu$ | N/A | N/A | N/A | 0.02 | 0.02 | 0.02 | 0.02 |
| Temperature $\alpha$ | N/A | N/A | N/A | N/A | 0.001 | N/A | 0.001 |
| Projection Step $\eta_a$ | N/A | N/A | N/A | N/A | N/A | 1E-4 | 1E-4 |
| Max GRG Updates $K$ | N/A | N/A | N/A | N/A | N/A | 10 | 10 |
| Projection Step in Evaluation $\eta_a^e$ | N/A | N/A | N/A | N/A | N/A | 1E-4 | 1E-4 |
| Max GRG Updates in Evaluation $K^e$ | N/A | N/A | N/A | N/A | N/A | 50 | 50 |

Table 6: Hyper-parameters for experiments in OPF with Battery Energy Storage.

would like to clarify that the principal purpose of the systematical action division method is to cope with the problem that $J_{:,m+1:n}^F$ is not invertible under some division or the equations defined by basic actions are not solvable. In that case, we will separate the situation into situations with linear equality constraint and nonlinear equality constraints, and illustrate different problems and corresponding systematical action division methods in both of them.

**Linear.** In the linear case, if $J_{:,m+1:n}^F$ is not invertible, this may result from two possible problems. Firstly, it indicates there exist redundant equality constraints, then we can directly delete these redundant equality constraints (i.e., choose the maximal linearly independent subset of row vectors of the coefficient matrix) and redefine the dimension of basic actions to make $J_{:,m+1:n}^F$ invertible. For example, if we have 3 linear equality constraints $Ax + b = 0$ on 4 actions, where

$$[A\,|b] = \begin{pmatrix} 1 & 0 & -2 & 3 & \bigg| & 2 \\ 5 & -3 & 1 & 4 & \bigg| & -1 \\ 4 & -3 & 3 & 1 & \bigg| & -3 \end{pmatrix}.$$

Here we choose $a_1$ as the basic action, and then $J_{:,m+1:n}^F = A_{:,2:4}$ is not invertible, since the third equality constraint can be represented by another two constraints. Hence, we can delete this redundant constraint and the new equality constraints are $\tilde{A} + \tilde{b} = 0$, where

$$\left[\tilde{A}\,\Big|\tilde{b}\right] = \begin{pmatrix} 1 & 0 & -2 & 3 & \bigg| & 2 \\ 5 & -3 & 1 & 4 & \bigg| & -1 \end{pmatrix}.$$

Then, the new basic actions will be $(a_1, a_2)$ and new $J_{:,m:n}^F = \tilde{A}_{:,2:4}$ is invertible. Another problem is with respect to the division of basic and nonbasic actions. For example, let us consider other 2 linear equality constraints $Ax + b = 0$ without redundant equality constraints on 3 actions as below:

$$[A\,|b] = \begin{pmatrix} 1 & -1 & -2 & \bigg| & 2 \\ 5 & -1 & -2 & \bigg| & -1 \end{pmatrix}.$$

Although there is no redundant equality constraint, $J_{:,m+1:n}^F$ will be still not invertible if $(a_2, a_3)$ are divided as nonbasic actions. Therefore, after removing the redundant equality constraints, we need to choose actions that correspond to the maximal linearly independent column vectors of the coefficient matrix as the nonbasic actions to ensure the invertibility of $J_{:,m+1:n}^F$.

**Nonlinear.** In the nonlinear case, the systematical way that divides the full action into the basic and nonbasic actions is principally to ensure the solvability of the equality equation.

Firstly, assume there are $n - m$ equality constraints and the full action $a \in \mathbb{R}^n$. We can construct a $0 - 1$ relationship matrix $E$ with the shape of $(n - m) \times n$, and $E_{ij}$ is to describe whether the equality constraint $f_i$ is related to $a_j$.

For example, if we have 3 equality constraints on action $a \in \mathbb{R}^4$ like that

$$f_1(a_1, a_2, a_4) = 0, f_2(a_2, a_3) = 0, f_3(a_1) = 0,$$

Then, the relationship matrix will be

$$\begin{pmatrix} 1 & 1 & 0 & 1 \\ 0 & 1 & 1 & 0 \\ 1 & 0 & 0 & 0 \end{pmatrix}$$

Now, we need to choose the nonbasic actions that cover the equality constraints as many as possible. This is the maximal matching problem in the bipartite graph, where constraints and actions are nodes and there exists an edge between constraint $f_i$ are related to action $a_j$ if $E_{ij} = 1$. Hence, $(a_1, a_2, a_3), (a_1, a_2, a_4), (a_1, a_3, a_4)$ are valid choices of nonbasic actions here, and then the equations can be solved with such divisions. In contrast, if we choose $(a_2, a_3, a_4)$ as the nonbasic actions, the equations cannot be solved. It is because if basic action $a_1$ is determined, the equations will be

$$f_1(a_2, a_4; a_1) = 0, f_2(a_2, a_3) = 0, f_3(\emptyset; a_1) = 0,$$

which is unsolvable.

Besides, $J^F_{:,m+1:n}$ may not be invertible in some specific action points. In such situation, we can add a small perturbation on the current action point or $J^F$ to make $J^F$ invertible. This is a useful trick in classical control methods. However, this situation is seldom seen in practice.

