# OpenReview forum: "Reduced Policy Optimization for Continuous Control with Hard Constraints"
_NeurIPS.cc/2023/Conference — NeurIPS 2023 poster_

### Official Review · Reviewer_DYt8 · 2023-06-25

**Soundness:** 3 good
**Presentation:** 3 good
**Contribution:** 2 fair
**Rating:** 6
**Confidence:** 4

**Summary:**

This paper proposes RPO to handle general hard constraints (equality and/or inequality constraints per step) for RL. The framework consists of construction and projection stages with a penalty loss for end-to-end training. Finally, the authors validate the effectiveness of their approach on 3 test benchmarks with hard constraints and compare them to previous CMDP-based approaches.

**Strengths:**

This paper studies safe RL with hard constraints which is an important problem. It is well-motivated and well-organized. The experiment results are promising as the proposed approach shows advantages against the CMDP baselines.  This paper is novel as the first attempt to introduce GRG to RL to solve the hard equality and inequality constraints.

**Weaknesses:**

There are several weak points of this paper for the reviewer.

1. This paper misses the recent literature addressing RL with hard safety constraints, such as

 Wang, Y., Zhan, S. S., Jiao, R., Wang, Z., Jin, W., Yang, Z., ... & Zhu, Q. (2022). Enforcing Hard Constraints with Soft Barriers: Safe Reinforcement Learning in Unknown Stochastic Environments. arXiv preprint arXiv:2209.15090.

This recent ICML paper also claims they are dealing with hard safety constraints for RL.

2. For the reviewer, the two stages proposed by the authors work like a shielding to modify the action generated by the policy network to satisfy the hard constraints. The reviewer would like to know how the authors compare their approach to the literature on RL + shielding, such as

Bastani, Osbert, Shuo Li, and Anton Xu. "Safe Reinforcement Learning via Statistical Model Predictive Shielding." Robotics: Science and Systems. 2021.

3. The MDP formulated in this paper assumes a general stochastic environment, this could raise a problem - "how could you let a random variable (s_t, a_t) be less than 0" for Eq(3), I feel it should be within the probability format, e.g., "Pr(g <= 0) = 1". The tested examples are all deterministic environments as continuous tasks. Therefore, the authors may want to reconsider how to formulate their safe RL problem.



**Questions:**

1. What are the soft constraints mentioned in the introduction?

2. The reviewer cannot agree that "it is worth noting that there is currently a lack of RL environments with complex hard constraints". The reviewer feels you can always define your own constraints either hard or soft as long as you have the environment model and full observability. The reviewer acknowledges the efforts to develop test examples but cannot agree those three can be claimed as "benchmarks"

3. Even if the final action satisfies the constraints, how should I know it is because of the two-stage procedure or the training objective with penalty loss?

4. How to enforce the policy network output basic actions?

**Limitations:**

The authors have clearly presented the limitations in the paper.

---

> ### Author Rebuttal · Authors · 2023-08-09
>
> We thank the reviewer for his/her careful reading and valuable suggestions. Below we will answer your concerns point-by-point.
>
> > **Q1**: This paper misses the recent literature [R5], which also deals with hard safety constraints for RL.
>
> **A1**: Thank you for your valuable suggestion. We have carefully studied [R5] and will cite it in our final version. However, the studied problem in [R5] is different from that in our works. As the authors illustrated in this paper, they attempt to solve the chance-constrained RL problem, which has the form:
> $$
> \begin{array}{c}
> \max_{\theta} J\left(\pi_{\theta}\right), \\\\
> \text{ s.t. }P\left(s(t) \notin S_{u}\mid\pi_{\theta}, s(0)\right)\geq 1-\eta,\forall t \in[0, T],\forall s(0)\in S_{0},
> \end{array}
> $$
> while the problem we studied is:
> $$
> \begin{aligned}
>         \max_\theta \ &J_R(\pi_\theta) \\\\
>         \text{ s.t. }\ &f_{i}(\pi_\theta(s_t);s_t)=0\quad\forall i, t, \\\\
>        &g_{j}(\pi_\theta(s_t); s_t)\leq 0\quad\forall j, t,
> \end{aligned}
> $$
> Although [R5] can satisfy the safety chance constraints, the safety chance constraints themselves may still provide a soft guarantee for the satisfaction of the actual constraints.
>
> [R5] Wang, Yixuan, et al. "Enforcing hard constraints with soft barriers: Safe reinforcement learning in unknown stochastic environments." ICML 2023.
>
> > **Q2**: The reviewer would like to know how the authors compare their approach to the literature on RL + shielding, such as [R6].
>
> **A2**: Thank you for raising the concern. We think the RL + shielding method described in [R6] also differs from our RPO. There are at least two different points. Firstly, the RL + shielding method in [R6] does not perform direct modification on the action generated by the policy network like our RPO but maintains two policies. One is for maximizing the cumulative reward, and another backup policy will be switched to when the potential violation exists. We think such methods should be classified as the methods based on Recovery like [40], which we have mentioned in line 90.
>
> Secondly, the problem studied by [R6], is also different from our RPO. RPO aims to solve hard instantaneous constraints, which are commonly very tight and exist in all states. For example, we must always satisfy the power balance equations in smart grid control. However, the RL + shielding method in this paper can only deal with the long-term soft constraints, which are not very tight, and violations may only exist in a few states. Hence, it is hard for this kind of method to solve RL with the hard constraints we studied.
>
> [R6] Bastani, O., Li, S., & Xu, A. (2021). Safe Reinforcement Learning via Statistical Model Predictive Shielding. In Robotics: Science and Systems (pp. 1-13).
>
> > **Q3**: The MDP formulated in this paper assumes a general stochastic environment; this could raise a problem - "how could you let a random variable (s_t, a_t) be less than 0" for Eq(3), I feel it should be within the probability format. Therefore, the authors may want to reconsider how to formulate their safe RL problem.
>
> **A3**: Thank you for raising the concern. As we explained in **A1, A2**, our RPO never aims to solve the Safe RL problem but attempts to solve MDP with hard constraints. In contrast to Safe RL methods, RPO focuses on decision problems with explicit instantaneous constraints like smart grid control. The difficulties of our studied problems are mainly from the tightness of the equality constraints and the requirement to maintain the satisfaction of both equality and inequality constraints rather than the randomness in constraints. For example, if the power balance constraints in the smart grid are modelled in probability form, it may lead to power unbalance in some states, and that will ruin the whole electrical grid.
>
> > **Q4**: What are the soft constraints mentioned in the introduction?
>
> **A4**: The soft constraints we mentioned are the constraints that are merely probably satisfied or satisfied with the expectation form. These kinds of constraints are more like multiple objectives, such as resource limitations. The soft constraint is different from the hard one. In some real-world applications like the smart grid, the hard constraints indicate the feasibility, which must be satisfied with high precision.
>
> > **Q5**: The reviewer cannot agree that "... there is currently a lack of RL environments with complex hard constraints". The reviewer acknowledges the efforts to develop test examples but cannot agree those three can be claimed as "benchmarks."
>
> **A5**: Thank you for raising the concern. More precisely, most existing benchmarks only consider inequality constraints, lacking RL environments with both hard equality and inequality constraints at present. Besides, the general interface needs to be carefully designed that delivers explicit information on constraints to the RL agent. Hence, there is currently a lack of RL environments with complex hard constraints. That's why we believe that our three environments can be claimed as "benchmarks" for the further study of RL with hard constraints.
>
> > **Q6**: Even if the final action satisfies the constraints, how should I know it is because of the two-stage procedure or the training objective with penalty loss?
>
> **A6**: Thank you for raising the concern. As lines 258-260, SAC-L and DDPG-L are the versions of RPO without the two-stage procedure. You can find in Figure 3 and Table 2 that their performance of them is much poorer than RPO-SAC and RPO-DDPG.
>
> > **Q7**: How to enforce the policy network output basic actions?
>
> **A7**: The actions are divided into basic and nonbasic actions before the training, and the division is not changed after that. Therefore, we do not need to enforce the policy network output basic actions but define the outputs of the policy network are the basic actions. More details on division can be referred to in our response **A1** to reviewer ixcf and **Global Response A2**.

---

> > ### Comment · Reviewer_DYt8 · 2023-08-12
> >
> > Thanks for the rebuttals, which address most of my questions.
> >
> > However, at least there is a formal math writing problem.
> > it is not clear to me that whether the authors are solving the safe RL problem in a general continuous and stochastic MDP (line 119 - 130) or a deterministic continuous environment where you can write (f(s_t, a_t) = 0, g(s_t, a_t) <= 0).  Please keep in mind that line 119 - 130 essentially assumes that s_t, a_t are random variables, to compare a random variable to a number (e.g., 0), you have to either use Expectation operator or Probability operator.
> >
> > My understanding is that this paper only deals with deterministic environment (because you want to write equality and inequality constraints directly on state and action without E[] and P()), you should have problem formulated as
> > s_{t+1} = f(s_t, a_t), f is unknown and continuous,  rather than the transition probability function P : S × A × S → [0, 1]

---

> > > ### Author Response · Authors · 2023-08-15
> > > **Response**
> > >
> > > Thanks for your valuable suggestion as well. We are glad that our response helps you solve most of your questions and will present a mathematical formulation of MDP with hard constraints more formally in the final version. Here are some further interpretations for your concerns.
> > >
> > > First, we need to clarify that we are studying RL with hard deterministic constraints in a general continuous and stochastic MDP. To further illustrate the properties of such a problem and present a formal mathematical formulation, we will show the differences between the standard Constrained MDP (CMDP) which is often studied by Safe RL methods, and our targeted problem --- MDP with hard constraints.
> > > - CMDP [RR1] introduces an extra function $C: S\times A\times S \rightarrow \mathbb{R}$ based on MDP (line 119-130). The constraints are viewed as the form of discounted cumulative constraints, which are defined as $J_{C_{i}}(\pi)=\mathbb{E}\_{\tau \sim \pi} \left[\sum\_{t=0}^{\infty} \gamma^{t} C_i\left(s_{t}, a_{t}, s_{t+1}\right)\right]$. We denote the set of feasible stationary policies for a CMDP problem as $\Pi_{C} = \left\\{\pi \in \Pi: \forall i, J_{C_{i}}(\pi) \leq d_{i}\right\\}$. The goal of safe RL is to find an optimal policy $\pi$  that maximizes the discounted cumulative reward in the set of feasible stationary policies $\Pi_{C}$.
> > >
> > > - **MDP with hard (deterministic) constraints** incorporates hard equality constraints and inequality constraints into the standard MDP. Concretely, the set of stationary policies that satisfy the hard equality constraints and inequality constraints are denoted as $\Pi_{F} =\left\\{\pi \in \Pi: \forall i, \forall s, f_{i}\left(\pi(s); s\right)=0\right\\}$ and $\Pi_{G} =\left\\{\pi \in \Pi: \forall j, \forall s, g_{j}\left(\pi(s); s\right) \leq 0\right\\}$, respectively. **Notably, here $f_i$ and $g_j$ are the hard deterministic constraint functions while $s_t$ and $a_t$ still involve randomness.**
> > >
> > > You can find that our formulation of MDP with hard constraints is actually a special case of CMDP and it focuses on the hard instantaneous and deterministic constraints. Actually, Our test environment of OPF with battery energy storage has a Gaussian perturbation in its transition function, and the results on it indicate that RPO has the capability to deal with the randomness in transition.
> > >
> > > In other words, the deterministic part in our problem formulation is only the constraint function and does not cover the transition function.
> > >
> > >
> > > [RR1] Altman, E. (1999). Constrained Markov decision processes (Vol. 7). CRC press.

---

> > > > ### Comment · Reviewer_DYt8 · 2023-08-15
> > > > **Soundness Issue of this Paper**
> > > >
> > > > I am pretty sure that we **cannot** write a deterministic constraint function on random variables without expectation $\mathbb{E}[]$ or probability $Pr()$ operators. Therefore, the definition $\Pi_F = \{ \pi \in \Pi: \forall i, \forall s, f_i(\pi(s); s) = 0\}$ is **wrong**.  The popular CMDP formulation is correct because the constraints are on the expectation of the random variables.
> > > > Given the paper is established on this **wrong** formulation, I am suspicious abouy the soundness of this paper. Therefore, I incline to decrease my score and raise this concern to other reviewers and AC.
> > > >
> > > > Please correct me if I am wrong, and I will recover the score.

---

> > > > > ### Author Response · Authors · 2023-08-16
> > > > > **Further illustrations of the constraint functions ---- Reply to Reviewer DYt8**
> > > > >
> > > > > Thanks for your feedback. We believe the formulation of the constraint $f(s_t, a_t)=0$ and $g(s_t, a_t)\leq 0$ without E[] or Pr() is not wrong. **The $s_t$ and $a_t$ in our instantaneous constraints are actually deterministic**. We are afraid that the claim in the last response "$s_t$ and $a_t$  contain randomness" may mislead you. Here we provide illustrations as follows.
> > > > >
> > > > > At the current time step $t$, $s_t$ is deterministic since $s_t$ has been observed, and $a_t$ is also deterministic as the deployed policy is deterministic. Our formulation considers instantaneous constraints at the current time step $t$ as well, therefore, the constraints are also deterministic. Thus we use $f(s_t, a_t)=0$ and $g(s_t, a_t)\leq 0$ without the Expectation Operator E[] or Probability Operator Pr().
> > > > >
> > > > > Actually, our claim "$s_t$ and $a_t$ contain randomness" in the last reply is from the perspective of the transition probability function. The transition probability function used in this paper (as shown in Line 121-122) is $P(s_t|s_{t-1}, a_{t-1})$. Here $s_{t-1}, a_{t-1}$ are the fixed conditions, while $s_t$ is the random variable. In other words, "$s_t$ and $a_t$ contain randomness" is conditioned on $(s_{t-1},a_{t-1})$.
> > > > >
> > > > > In a standard constrained MDP (CMDP), the constraints involve both $(s_t, a_t)$ and the **random** next state $s_{t+1}$. Therefore, the expectation or probability operations must be used in the constraint functions of CMDP. Our instantaneous constraint functions are with respect to only $(s_t, a_t)$. That's why we don't use such operators.
> > > > >
> > > > > Similar constraint formulation in RL without E[] or Pr() can also be referred to in [RRR1].
> > > > >
> > > > > We hope the above clarification can eliminate your concerns. Please feel free to let us know if you have further concerns.
> > > > >
> > > > > [RRR1] Liu, P., Tateo, D., Ammar, H. B., & Peters, J. (2022, January). Robot reinforcement learning on the constraint manifold. In Conference on Robot Learning (pp. 1357-1366). PMLR.

---

> > > > > > ### Comment · Reviewer_DYt8 · 2023-08-17
> > > > > > **Responses Addressed My Soundness Concerns**
> > > > > >
> > > > > > The responses make sense to me and addressed my soundness concern.
> > > > > > Please make it clear that what are the random variables, what are the realization/observation of random variables, and how the instantaneous constraints formulated and differ from CMDP in the paper revision.
> > > > > > I will recover my score to the original 6.
> > > > > > Thanks for the responses.

---

> > > > > > > ### Author Response · Authors · 2023-08-20
> > > > > > >
> > > > > > > Thanks for recovering your score. We will present clearer illustrations about the problem formulation in our paper revision accordingly.

---

### Official Review · Reviewer_u5yD · 2023-06-27

**Soundness:** 3 good
**Presentation:** 3 good
**Contribution:** 3 good
**Rating:** 7
**Confidence:** 3

**Summary:**

Inspired by the GRG algorithm, this paper proposed a new reduced policy optimization (RPO) algorithm to handle hard equality and inequality constraints that must be satisfied by any learned policies for continuous control. The algorithm consists of two separate phases. Phase 1 involves the training of the policy network and the equation solver for tackling the equality constraints. Phase 2 further utilizes GRG to tackle the remaining inequality constraints without violating any equality constraints. Experiments conducted on three benchmark problems show that RPO allows the trained policies to achieve high cumulative rewards while minimizing the potential chances of constraint violation, compared to several baseline algorithms.

**Strengths:**

It is important to develop new learning algorithms that can handle a variety of hard constraints requested for solving a reinforcement learning problem. This paper proposed an interesting new algorithm design towards achieving this goal. The validity of some aspects of the new algorithm design is also supported by some theoretical results. Meanwhile, the experiment results look promising.

**Weaknesses:**

I have some concerns regarding the novelty of the constrained reinforcement learning problems studied in this paper. Past research works often studied constraints regarding the expected behaviors of the trained policy across a full episode. In other words, the constraints are defined over a long sequence of states and actions where all state-action pairs in the sequence are interdependent. In contrast, this paper considered constraints that can be independently defined for every state-action pair. While this consideration certainly has its practical value, however the RPO algorithm developed in this paper may not be directly comparable to those algorithms proposed for solving the time correlated constraints rather than the time independent constraints. Hence, the real technical contribution of RPO for handling multiple equality and inequality constraints may need to be more precisely described and clearly justified.

The experimental evaluation was performed on three relatively simple reinforcement learning problems. To fully understand the technical advantages of the new RPO algorithm over existing approaches, it may be important to conduct further evaluations of the RPO algorithm on more (and perhaps more challenging) benchmark problems. For example, the review of existing hard-constrained reinforcement learning algorithms in Section 2 seems to suggest a few additional benchmark problems that may need to be considered in this paper too.

Other than GRG, there are a variety of ways to handle time independent constraints on state-action pairs. While this paper introduced the basic idea of GRG clearly, the importance of using GRG over other competing constraint handling or constrained optimization methods may need to be further clarified and justified. Additional theoretical and empirical evaluations may be helpful to clearly reveal the technical novelty of using GRG for constrained policy optimization.

This paper seems to assume that equality constraints can be easily satisfied by separating all dimensions of the action space into basic dimensions and nonbasic dimensions. Upon fixing the basic dimensions, it is feasible to find suitable values for the nonbasic dimensions to satisfy all equality constraints. This assumption may be debatable and may need to be further investigated. Particularly, it is not clear to me how to systematically divide the full action space into the basic and nonbasic subspaces. Will any arbitrary division work? What will be the impact of any division on the performance of the learned policy? Furthermore, given that the equality constraints can be highly sophisticated and nonlinear, it may not always be possible for the equation solver to find feasible action that can satisfy all equality constraints, subject to the division used.

What seems also questionable regarding the new algorithm design is the necessity of using Lagrangian penalties in the loss function for training policy networks. I understand that this may make the projection stage more efficient and reliable. However, the penalties do not seem compulsory and are subject to the complexity of the inequality constraints. For the benchmark problems studied in the experiments, it seems that the inequality constraints are not hard to satisfy. The authors mentioned some existing approaches that can satisfy the inequality constraints directly. In view of this, the necessity of using Lagrangian penalties in the loss function should be further studied, both theoretically and experimentally. Meanwhile, the authors may need to evaluate their RPO algorithm on benchmark problems with more challenging inequality constraints, in order to better demonstrate its strong constraint handling capabilities.

Besides the above, according to my understanding, the projection stage is treated as part of the learning environment. Does this stage depend fully on the action output from the construction stage? If the projection stage is random or time-varying, the stability of the policy network training process may be affected. Furthermore, the reparameterization trick requires propagating the gradients back from the action space to the trainable parameters in the policy network. The effectiveness of this trick may be affected since the projection stage may not be easily supported by the gradient calculation map. On the other hand, if the learning environment is re-defined such that the projection stage is treated as part of the environment, this trick may also affect the effectiveness and reliability of the policy network training process and may need to be further examined.

Since the benchmark problems studied in this paper are new, please give a more detailed explanation of each problem in this paper. For example, the authors mentioned that the equality constraint of the Spring Pendulum problem is state-dependent. However, the actual dependence relationship is not presented, making it hard to understand how the equality constraint is defined and whether it is difficult to satisfy the equality constraint.

**Questions:**

What is the real technical contribution of RPO for handling multiple time independent equality and inequality constraints?

How to systematically divide the full action space into the basic and nonbasic subspaces? Will any arbitrary division work? What will be the impact of any division on the performance of the learned policy?

When is it important and necessary to use Lagrangian penalties in the loss function for training policy networks?

**Limitations:**

I do not have any concerns regarding this question.

---

> ### Author Rebuttal · Authors · 2023-08-09
>
> We thank you for your thorough reviews to help us improve the quality of our work. We will answer all the questions that you concern about.
>
> > **Q1**: What is the real technical contribution of RPO?
>
> **A1**: Thank you for your constructive suggestion. The real technical contribution of RPO is to handle both hard instantaneous (time-independent) equality and inequality constraints in a general RL paradigm. It's worth noting that past research works in constrained RL cannot handle such hard instantaneous constraints. The properties of RPO and the differences between RPO and other methods have been summarized in Table 1. In the final version, we will present the technical contribution of RPO more precisely.
>
> > **Q2**: The experimental evaluation was performed on three relatively simple problems ... additional benchmark problems may be needed.
>
> **A2**: We respectively disagree with this comment that our test problems are relatively simple. Please refer to the **Global Response A1**.
>
> > **Q3**: The importance of using GRG over other competing constraint handling or constrained optimization methods may need to be further clarified and justified. Additional theoretical and empirical evaluations may be added for GRG.
>
> **A3**: Thank you for your constructive suggestion. As we illustrated in **A1**, RPO utilizing the GRG method can handle both general hard instantaneous equality and inequality constraints, while previous research approaches cannot handle this problem.
>
> With respect to additional empirical evaluation, although there are various RL methods for time-independent constraints, most of them only consider inequality constraints. Hence, they will fail in our benchmarks with both equality constraints and inequality constraints like Safety Layer [14]. This has been shown in our experiments in Table 2 and Figure 3.
>
> With respect to additional theoretical evaluation, we think the constraint satisfaction of RPO is from GRG updates post-plugged in the policy network, while previous constrained RL methods directly use actions from the policy network and enforce the constraint satisfaction via optimization tricks during training. Hence, the theoretical guarantee of RPO on constraint satisfaction is obvious since the convergence of GRG and Newton's method has been proved in many classical optimization books [25]. Instead, we have presented two theorems to show the property of our RPO, including the GRG update executed in the tangent space and the exact penalty theorem for inequality constraints.
>
> > **Q4**: How to divide the full action space into the basic and nonbasic subspaces? What will be the impact of division?
>
> **A4**: Thank you for raising the concern. If you meet up with cases where there exists an equality constraint that is not related to the full action space. The systematical way can be referred to as our **Global Response A2**.
>
> > **Q5(1)**: Some existing approaches can satisfy the inequality constraints directly; the necessity of using Lagrangian penalties in the loss function should be studied. When is it important and necessary to use this penalties?
>
> **A5(1)**: Thank you for your valuable suggestion. However, we cannot agree with you that the Lagrangian penalties are unnecessary. Here are two reasons. Firstly, if we remove the exact penalty term in the loss function, the constraint violation of $\tilde{a}$ will be very large, which will lead to thousands of iterations in the projection stage. It is obviously unacceptable in real-world applications. Secondly, if there exists a large gap between the initial action $\tilde{a}$ and the final action $a$, training without the Lagrangian penalties will be unstable.
>
> In addition, while we mentioned some existing approaches that can satisfy the inequality constraints directly, most of them cannot deal with equality constraints, like Safety Layer [14], as we showed in Table 2 and Figure 3.
>
> > **Q5(2)**: It seems that the inequality constraints are not hard to satisfy for the benchmark problems. Require challenging inequality constraints ...
>
> **A5(2)**: Please refer to our **Global Response A1(2)**.
>
> > **Q6(1)**: Does the projection stage depend fully on the action output from the construction stage? Is the projection stage random or time-varying?
>
> **A6(1)**: Yes, the projection stage depends fully on the action output from the construction stage without any extra randomness. However, during the training process, this output action of the given state changes, which may lead to a changing number of iterations in the projection stage. Hence, we do not let the projection stage participate in the backpropagation, as shown in Equation 8.
>
> > **Q6(2)**: Treating the projection stage as part of learning may affect the effectiveness and reliability of the policy network.
>
> **A6(2)**: The projection stage is not used as a part of the learning environment. Instead, as shown in Algorithm 1, it is used in the TD target since the action output by the projection stage is the actual behaviour policy. As we analyzed in lines 237-247, that's why we chose off-policy RL methods as the base of RPO to ensure the stability of the training.
>
> > **Q7**: A more detailed explanation of each problem. Why the equality constraint of the Spring Pendulum is state-dependent?
>
> **A7**: Thank you for your constructive suggestion. We will add a more detailed explanation of our benchmark problems in the final version. Below we will explain why the equality constraints in Spring Pendulum are state-dependent. As shown in Appendix D.2, the equality constraint of the spring pendulum is $\dot{l}+\ddot{l}dt=0$, where $\dot{l}$ is the stretching velocity of the spring and $\ddot{l}$ is the acceleration of the spring. According to Euler-Lagrange equations, we have
> $$
> \ddot{l}=\frac{f_{s}+m l \theta^{2}-k\left(l-l_{0}\right)-mg\cos\theta}{m}.
> $$
> This means the equality constraint $\dot{l}+\ddot{l}dt=0$ is related to the state variables: the cosine and sine of the angle.

---

> ### Comment · Reviewer_u5yD · 2023-08-11
> **Thank the authors for their response**
>
> I would like to thank the authors for their response to my concerns. The response addressed some of my concerns. I still don't quite understand why it is very important to handle time-independent constraints for many real-world applications. Furthermore, if the projection stage is effective at handling constraints, I don't understand why we should worry about the Lagrangian penalties while training the policy networks. This seems to suggest that the projection stage may fail to satisfy the time-independent constraints of the problem being solved. I guess more empirical and theoretical analysis regarding this issue may be helpful. However, I am more clearer about the technical details based on the authors' response and will raise my recommendation a bit.

---

> > ### Author Response · Authors · 2023-08-15
> > **Response**
> >
> > Thank you for the careful and thorough reviews. We are glad that our response addressed some of your concerns about the technical details. We will provide further clarification for your concerns below and add corresponding analysis in our final version.
> >
> > **(1)**: Why it is very important to handle time-independent constraints for many real-world applications?
> >
> > Here we need to clarify 2 points:
> >
> > - **Some real-world applications do have such hard time-independent (instantaneous) constraints.** For example, the time-independent power flow constraints and voltage-bound constraints in the smart grid should be fulfilled at each time slot. If these constraints are not satisfied exactly, we cannot deploy RL method there. Although existing Safe RL algorithms claim they can handle both time-correlated and time-independent constraints, they just satisfy the constraints with certain probability or with expectation form. This means that they are not applicable to the applications mentioned above; our experiments shown in Figure 3 and Table 2 also confirm that.
> > Therefore, it is important to design RPO, which is customized for handling hard instantaneous constraints in some applications.
> >
> > - Our RPO can be viewed as **a supplement to Safe RL algorithms**, since there is no contradiction to using RPO and one of most Safe RL algorithms like CPO at the same time. If we meet up with a real-world application with both hard instantaneous and soft cumulative (time-correlated) constraints, **we can use RPO to handle the hard instantaneous constraints and meanwhile apply some existing Safe RL method to handle soft cumulative constraints.**
> >
> > **(2)**: Why do we need to use the projection stage and Lagrangian penalties at the same time when dealing with inequality constraints?
> >
> > We would like to clarify that handling hard inequality constraints exactly in any state is challenging, especially together with hard equality constraints. Hence, it is not enough to use only either the projection stage or Lagrangian penalties. Here are concrete reasons from two aspects.
> >
> > - If we only use the projection stage to handle inequality constraints, the training process will be **unstable and inefficient**. Using Lagrangian penalties in the loss can generate "good" initial actions for the projection stage.
> >
> > - If we only use the Lagrangian penalties to handle inequality constraints, it cannot satisfy the inequality constraints exactly for **the approximation and generalization error of policy network**. As shown in Table 3, the inequality satisfaction of SAC-L and DDPG-L is poorer than RPO-SAC and RPO-DDPG. The projection stage can further reduce the constraint violation.
> >
> > Therefore, we need to use both the projection stage and Lagrangian penalties at the same time when dealing with inequality constraints.

---

> > > ### Comment · Reviewer_u5yD · 2023-08-18
> > > **Thank you for the further clarification**
> > >
> > > I would like to thank the authors for the further clarification. Based on that, I will further increase my rating a bit.

---

### Official Review · Reviewer_ixcf · 2023-07-03

**Soundness:** 3 good
**Presentation:** 3 good
**Contribution:** 3 good
**Rating:** 6
**Confidence:** 3

**Summary:**

The paper solves the RL problem with equality and inequality hard constraints with a reduced policy optimization (RPO) algorithm, which combines RL with the generalized reduced gradient (GRG) algorithm. RPO partitions actions into basic actions and nonbasic actions following the GRG method, outputs the basic actions using a policy network, and calculates the nonbasic actions using domain knowledge to satisfy equality constraints. Then, the actions go through a projection stage to handle inequality constraints. Experimental results show that the proposed algorithm behaves better than the baselines for MDP with hard constraints.

**Strengths:**

Considering both equality and inequality constraints in RL is an interesting problem. The proposed algorithm can handle the problem well. Also, the paper introduced some new RL benchmarks with hard constraints.


**Weaknesses:**

1. In Equation (4), it seems that there is an implicit assumption that $J^F_{\:,m\:n}$ is invertible. However, this may not always be true. I think in some environments, the equality constraints may even do not depend on the actions but only the states. For example, if we want a car to always stay in a pre-defined path, then $F$ is only a function of states, and $\frac{\partial F}{\partial a}=0$. The proposed approach seems cannot solve this problem.

2. In the experiments, the equality constraints can be solved by constructions, e.g., in Figure 2(a), let $f_2=\frac{f_1 \cos\theta_1}{\cos\theta_2}$, where $\theta_1$/$\theta_2$ are the angles between $f_1$/$f_2$ and the vertical direction. It is unclear to me why we need to satisfy the equality constraints using 4.1 instead of just satisfying them by construction since the domain knowledge required by 4.1 and by construction are the same.

3. It seems that there is a mismatch between Figure 3 and Table 2, where the constraint violation of RPO methods are less than $10^{-4}$ in the first two environments in Table 2, but can be larger than $10^{-4}$ in Figure 3. It is also not clear about the meaning of “max violation”.

4. There are typos even in the mathematical parts of the paper. For example, line 161 $a^N\in\mathbb{R}^m$ -> $a^N\in\mathbb{R}^{n-m}$; line 173 $J^F\in\mathbb{R}^{(m-n)\times n}$ -> $J^F\in\mathbb{R}^{(n-m)\times n}$.

5. Constraint violation still exists in the experiments, and there is a lack of discussion.


**Questions:**

1. Do you assume that  $J_{:,m:n}^F$ is invertible in Equation 4? How can we handle the case when $F$ is only a function of states?

2. Why do we need to satisfy the equality constraints using 4.1 instead of just satisfying them by construction since the domain knowledge required by 4.1 and by construction are the same (the second point in Weaknesses)?

3. Why is there a mismatch between Figure 3 and Table 2 (the third point in Weaknesses)?

4. Can the authors give an analysis of the reason why constraint violation still exists in the experiments?


**Limitations:**

The authors address the limitations well in Section 6.

---

> ### Author Rebuttal · Authors · 2023-08-09
>
> We thank the reviewer for the constructive feedback on our paper. Here we address your detailed questions as below:
>
> > **Q1**: In Equation (4), it seems that there is an implicit assumption that $J^F_{:, m:n}$ is invertible. However, this may not always be true.
>
> **A1**: Thank you for raising the concern. We will analyze the invertibility of $J^F_{:, m:n}$ in two situations: linear equality constraints and nonlinear equality constraints.
>
> In the linear case, if $J^F$ is not invertible, it means there exist redundant equality constraints, then we can directly delete these redundant equality constraints and redefine the dimension of basic actions to make $J^F$ invertible. For example, if we have 3 linear equality constraints $Ax+b=0$ on 4 actions, where
> $$
> \left[A\left|b\right.\right]=\left(
> \begin{array}{cccc|c}
>     1 & 0 & -2 & 3 & 2\\\\
>     5 & -3 & 1 & 4 & -1\\\\
>     4 & -3 & 3 & 1 & -3
> \end{array}\right).
> $$
>
> Here we choose $a_1$ as the basic action, and then $J^F_{:, m:n}=A_{:, 1:4}$ is not invertible, since the third equality constraint can be represented by another two constraints. Hence, we can delete this redundant constraint and the new equality constraints are $\tilde{A} + \tilde{b} = 0$, where
> $$
> \left[\tilde{A}\left|\tilde{b}\right.\right]=\left(
> \begin{array}{cccc|c}
>     1 & 0 & -2 & 3 & 2\\\\
>     5 & -3 & 1 & 4 & -1
> \end{array}\right).
> $$
>
> Then, the new basic actions will be $(a_1, a_2)$ and new $J^F_{:, m:n}=\tilde{A}_{:, 2:4}$ is invertible.
>
> In the nonlinear case, $J^F$ may not be invertible in some specific action points. In such situation, we can add a small perturbation on the current action point or $J^F$ to make $J^F$ invertible. This is a useful trick in classical control methods. Actually, this situation is seldom seen in practice.
>
> > **Q2**:  I think in some environments, the equality constraints may even do not depend on the actions but only the states. For example, if we want a car to always stay in a pre-defined path, then $F$ is only a function of states, and $\frac{\partial F}{\partial a}=0$. The proposed approach seems cannot solve this problem.
>
> **A2**: We believe our method can still handle the above problem by modifying the model formulation. Note that the current state $s_t$ is inherently defined by the last state $s_{t-1}$ and last action $a_{t-1}$, i.e. $s_t = f(s_{t-1},a_{t-1})$. Therefore, we can transform action-independent equality constraints into action-dependent ones. For example, if we want a car to always stay in pre-defined path, we can formulate the equality constraints on current action by predicting the next position (state).
>
> > **Q3**: It is unclear to me why we need to satisfy the equality constraints using 4.1 instead of just satisfying them by construction since the domain knowledge required by 4.1 and by construction are the same.
>
> **A3**: Thank you for raising the concern. Section 4.1 actually introduces how to backpropagate the gradient in the construction stage. If we satisfy equality constraints by direct construction and ignore the gradient $\frac{\partial\mathcal{L}}{\partial a^N}  \frac{\partial a^N}{\partial a^B}$ from the Construction Stage in Section 4.1, we will only obtain the partial gradient $\frac{\partial \mathcal{L}}{\partial a^B}$ on the basic actions as shown below. This will ruin the training of the policy network. The complete gradient flow is as follows,
> $$
> \nabla_{a^B} \mathcal{L} = \frac{\partial \mathcal{L}}{\partial a^B} + \frac{\partial\mathcal{L}}{\partial a^N}  \frac{\partial a^N}{\partial a^B}
> $$
>
> In Appendix E.3, we have also analyzed your concern with experiments. You can find in Figure 7 that RPO with only a partial gradient (without the gradient from construction) performs much worse than RPO with a gradient from construction in cumulative reward. In other words, if we construct the nonbasic actions directly and do not consider them in backpropagation, the performance of RPO will be very poor.
>
> > **Q4**: It seems that there is a mismatch between Figure 3 and Table 2, where the constraint violation of RPO methods are less than $10^{-4}$ in the first two environments in Table 2 but can be larger than $10^{-4}$ in Figure 3. It is also not clear about the meaning of “max violation”.
>
> **A4**: The mismatch between Figure 3 and Table 2 is because the former is for the training process while the latter is for the evaluation process. Here are two reasons why the evaluation performance is better than the training one. Generally, randomness is introduced during the training process for exploration while it is removed in the evaluation process. This trick is commonly seen in RL methods. Besides, we use more maximum GRG updates in the evaluation procedure. This is a practical trick that we have mentioned in Appendix E.1 (lines 625-627).
>
> The "max violation" in Figure 3 means the maximum violation of all constraints in one state, as we illustrated in lines 292-294.
>
> > **Q5**: There are typos even in the mathematical parts of the paper. For example, line 161 $a^N\in\mathbb{R}^m$ -> $a^N\in\mathbb{R}^{n-m}$; line 173 $J^F\in\mathbb{R}^{(m-n)\times n}$ -> $J^F\in \mathbb{R}^{(n-m)\times n}$.
>
> **A5**: Thank you for pointing out our typos. We will correct them in the final version.
>
> > **Q6**: Constraint violation still exists in the experiments, and there is a lack of discussion.
>
> **A6**: Thank you for your constructive suggestion. Our method involves multiple hard equality and inequality constraints. In the practical computation, the tolerance of the constraint satisfaction is usually set as 1e-3 or 1e-4. This is also achieved by our method. The constraint violation you see in the experiments is within this tolerance. we will add the discussion in Section 5.3 of the final version.

---

> > ### Comment · Reviewer_ixcf · 2023-08-14
> > **Response to the rebuttal**
> >
> > I would like to thank the authors for providing additional information and examples. The rebuttal addresses my concerns. Please modify the paper accordingly in the final version. I have raised my recommendation for the paper.

---

> > > ### Author Response · Authors · 2023-08-15
> > > **Response**
> > >
> > > Thanks for your positive feedback. We are glad that our response addresses your concerns and will modify the paper accordingly in the final version.

---

### Official Review · Reviewer_RHA4 · 2023-07-06

**Soundness:** 3 good
**Presentation:** 3 good
**Contribution:** 3 good
**Rating:** 6
**Confidence:** 4

**Summary:**

The authors introduce a policy optimization methodology suitable for continuous control problems with hard constraints. The optimization framework, named Reduced Policy Optimization (RPO), utilizes mathematical tools such as Generalized Reduced Gradient (GRG) and Lagrangian relaxation to address hard (equality) and soft (inequality) constraints respectively. Through GRG basic and non-basic actions are extracted. The policy network produces the basic actions and by solving the equality equations non-basic actions are calculated. The Lagrangian relaxation is implemented in the policy’s cost function to fulfill the inequality constraints. The evaluation of RPO occurred in three different environments the Safe Cartpole, Spring Pendulum, and OPF with battery energy storage.

**Strengths:**

- The proposed methodology is innovative since it introduces the GRG to the RL setting. The goal of RPO is to create a policy optimization with hard constraints in an agnostic manner, which can be very beneficial in the field of RL and set the agent capable to solve a variety of complex tasks.
- The manuscript is well-written and thorough. Related work and relative background are on point.
- The experiment results present significant improvements in comparison with current policy optimization methods.


**Weaknesses:**

- The number of experiments and the complexity is insufficient to back up the algorithm's robustness and versatility for complex tasks.
- To obtain the non-basic actions one should implement an equation solver depending on the task, which makes the implementation more difficult. How can we get define this solver in a generic way?
- In general, off-policy algorithms tend to be more time-consuming than on-policy algorithms. GRG adds more time complexity to the whole optimization process. The authors are aware of that. Hence a training time comparison should have been included.
- The included code contains only the cart pole benchmark.
- The code does not run as is: there is no version defined and a lot of deprecated numpy methods (not present on my machine for example) are used.


**Questions:**

- How do you think the RPO is probable to perform in complex tasks with hard constraints such as locomotion tasks or tasks requiring a robotic manipulator?
- What is the average training time using the RPO in comparison to vanilla policy optimization (e.g. SAC, DDPG)?
-  How can we get define this equation solver for the non-basic actions in a generic way?


**Limitations:**

The authors adequately discuss the limitations of their proposed method.

---

> ### Author Rebuttal · Authors · 2023-08-09
>
> We thank the reviewer for your valuable feedback and constructive comments! We itemize the weaknesses or comments you mentioned and answers to them.
>
> > **Q1**: The number of experiments and the complexity is insufficient to back up the algorithm's robustness and versatility for complex tasks.
>
> **A1**: Thanks for your comments. However, we cannot agree with you that our experiments are insufficient. We have tested our algorithm in three environments, and we noticed that similar papers like [R1, R2] tested 3-4 environments as well. Besides, the complexity of our three test environments has been demonstrated in **Global Response A1**.
>
> [R1] Liu P, Tateo D, Ammar H B, et al. Robot reinforcement learning on the constraint manifold[C]//Conference on Robot Learning. PMLR, 2022: 1357-1366.
>
> [R2] Wang, Y., Zhan, S. S., Jiao, R., Wang, Z., Jin, W., Yang, Z., … & Zhu, Q. (2023, July). Enforcing hard constraints with soft barriers: Safe reinforcement learning in unknown stochastic environments. In International Conference on Machine Learning (pp. 36593-36604). PMLR.
>
> > **Q2**: To obtain the non-basic actions one should implement an equation solver depending on the task, which makes the implementation more difficult. How can we get define this solver in a generic way?
>
> **A2**: This is a good question. Different from nonconvex optimization, there exist many efficient methods like modified Newton's method that can solve general nonlinear equations that possess global convergence. More theoretical analysis of the convergence of modified Newton's method can be seen in Chapter 10 of [R3]. Actually, in OPF with battery energy storage case, we apply Newton's method in the construction stage. Therefore, we can define the modified Newton's method as a general solver for this problem.   Notably, for some special equality constraints, we can use analytical solutions directly instead of the modified Newton's method. That's why we do not define a fixed solver.
>
> [R3] Luenberger D G, Ye Y. Linear and nonlinear programming[M]. Reading, MA: Addison-wesley, 1984.
>
> > **Q3**: The comparison of training time between GRG and off-policy algorithms should be included.
>
> **A3**: Thank you for your valuable suggestion. The training time comparison in the environment Safe CartPole and Spring Pendulum is shown in the following tables. Each result is averaged in 5 runs.
>
> For the time limitation, the training time comparison on OPF with battery energy storage is still in progress, and we will include all results in our final version.
>
> The training time comparison in Safe Cartpole.
>
> |  Method            |    RPO-DDPG   |     RPO-SAC   |    DDPG-L     |      SAC-L    |     CPO       |      CUP      | Safety Layer  |
> |:------------------:|:-------------:|:-------------:|:-------------:|:-------------:|:-------------:|:-------------:|:-------------:|
> | **Training Time (s)**  |   190.2    |    375.5  |          150.2     |        268.5      |      68.7        |   75.2  |      1067.0         |
>
> The training time comparison in Spring Pendulum.
> |  Method            |    RPO-DDPG   |     RPO-SAC   |    DDPG-L     |      SAC-L    |     CPO       |      CUP      | Safety Layer  |
> |:------------------:|:-------------:|:-------------:|:-------------:|:-------------:|:-------------:|:-------------:|:-------------:|
> | **Training Time (s)**  |   242.6    |    419.0  |      174.9    |        309.5      |     98.6      |   81.4  |    3859.7           |
>
>
> > **Q4 & Q5**: The included code contains only the cart pole benchmark. The code does not run as is: there is no version defined and a lot of deprecated numpy methods (not present on my machine for example) are used.
>
> **A4 & A5**: Thank you for raising the concern. According to the principle of NIPS 2023, we cannot present an anonymous repository during the rebuttal phase. Hence, we will release all codes in our three benchmarks later.
>
> To help you solve the code-running problems, we also present the versions of Python packages used in our work as below.
>
> ```
> numpy                         1.21.6
> torch                         1.13.1
> gym                           0.19.0
> matplotlib                    3.5.2
> scipy                         1.9.1
> scikit-learn                  1.0.2
> pandas                        1.4.4
> ```
>
> In addition, our experimental operating system is Ubuntu 20.04.6 LTS, the experimental GPU is NVIDIA GeForce RTX 3090 and the CUDA Version is 11.7 for your reference. If you still have any problems, please let us know and we will help you solve them to the experimental results.
>
> > **Q6**: How do you think the RPO is probable to perform in complex tasks with hard constraints such as locomotion tasks or tasks requiring a robotic manipulator?
>
> **A6**: Thank you for raising the concern. We believe that RPO can work in complex tasks with hard constraints. On one hand, as we illustrated in **A1**, our test environment --- OPF with battery energy storage, is actually a complex task with 28 equality constraints and 58 inequality constraints. In contrast, constraints in locomotion tasks or tasks requiring a robotic manipulator like [R4] are generally simpler than our OPF with energy storage. For example, [R4] only contains several constraints. On the other hand, our experiments in classical robot control such as cartpole can also confirm certain transferability of our method in robot control tasks.
>
> [R4] Liu, P., Tateo, D., Ammar, H. B., & Peters, J. (2022, January). Robot reinforcement learning on the constraint manifold. In Conference on Robot Learning (pp. 1357-1366). PMLR.

---

> > ### Comment · Reviewer_RHA4 · 2023-08-17
> > **Reviewer Response**
> >
> > I appreciate the thorough answers to my comments and the extra clarifications.
> >
> > Based on the training times provided, there is not much overhead due to the solver, which sets the RPO as a viable option when dealing with hard constraints. A discussion about the possible solvers and optimization tools could be included in the manuscript/supplementary.
> >
> > Additionally, the performance of your algorithm in the OPF with battery energy storage task, which consists of a large number of both equality and inequality constraints is a plus.
> >
> > Considering the above, I am increasing my score.

---

> > > ### Author Response · Authors · 2023-08-20
> > >
> > > Thanks for your further feedback and increasing the score. We will provide the corresponding discussion and the results of the additional experiments in the final version according to your suggestions.

---

### Author Rebuttal · Authors · 2023-08-09

We thank the reviewers for their careful reading of considerate and meaningful suggestions to help us improve our paper. We sincerely appreciate that the reviewers find our work "innovative" (RHA4), "interesting" (u5yD,  ixcf) and "novel and well-motivated as the first attempt to introduce GRG to RL" (DYt8) to RL with hard constraints, which can be "very beneficial" (RHA4) in the RL field, and can handle both equality and inequality constraints in "a variety of" "complex" tasks (RHA4, u5yD). We are further glad that the reviewers agree unanimously that our manuscript is "well-written and "thorough" (RHA4, DYt8) and confirm our contributions on both theoretical analysis and "promising" empirical results (RHA4, u5yD, DYt8) to support our algorithm, and "new" RL benchmarks with hard constraint (ixcf).

In the following, we will try to address the concerns/questions of the reviewers and present a detailed item-by-item response to their comments.

Firstly, we would like to offer several clarifications about the common issues from the reviewers.

**Q1**: **(1)** The three test environments used seem simple and insufficient to support the RPO algorithm, and additional benchmark problems should be considered. **(2)** The inequality constraints of three test benchmarks are simple, and more challenging inequality constraints need to be considered.

**A1**: **(1)** We believe our three test environments with both multiple equality and inequality constraints are complex enough to verify the efficiency of our algorithm empirically. For example, **the case of OPF with battery energy storage involves 57 states, 43 actions, and contains 28 equality constraints and 58 inequality constraints with high nonconvexity**. It's worth noting that there is a lack of such complex benchmarks to test our algorithm. We have to spend a lot of effort in developing these RL environments with both hard equality and inequality constraints. Current RL environments with hard constraints mentioned in Section 2 are either not open-source or require numerous modifications since they were designed just for one specific application. Implementing additional benchmark problems, such as UAVs, and robot dogs, is under consideration in our further work.

**(2)** We would like to highlight that there is **no need to evaluate our RPO with more challenging inequality constraints**. As we mentioned in lines 131-135, any nonlinear inequality constraints can always be transformed into equality constraints plus inequality box constraints by adding slack variables. Hence, evaluating RPO with box inequality constraints and nonlinear equality constraints is sufficient to show RPO's generality. For example, if we have a complex inequality constraint $g(s, a) \le 0$. Then, we can always transform it into
$$
    \begin{aligned}
        g(s,a) + \nu &= 0, \\\\
        \nu &\ge 0,
    \end{aligned}
$$
where $\nu$ is the slack variable, which can be viewed as the augmented actions in RL.

**Q2**: How to divide the action into basic and nonbasic actions? What's the impact of different division choices?

**A2**: The systematical way to divide the full action space into the basic and nonbasic subspaces below can ensure the solvability of the equality equation.

Firstly, assume there are $n-m$ equality constraints and the full action $a\in \mathbb{R}^n$. We can construct a $0-1$ relationship matrix $E$ with the shape of $(n-m)\times n$, and $E_{ij}$ is to describe whether the equality constraint $f_i$ is related to $a_j$.

For example, if we have 3 equality constraints on action $a\in \mathbb{R}^4$ like that
$$
\begin{aligned}
f_1(a_1,a_2, a_4)=0, f_2(a_2,a_3)=0,f_3(a_1)=0,
\end{aligned}
$$

Then, the relationship matrix will be
$$
\left(
\begin{array}{ccc}
1 & 1 & 0 & 1 \\\\
0 & 1 & 1 & 0 \\\\
1 & 0 & 0 & 0
\end{array}\right)
$$

Now, we need to choose the nonbasic actions that cover the equality constraints as many as possible. This is the maximal matching problem in the bipartite graph. Hence, (1,2,3), (1,2,4), (1,3,4) are valid choices of nonbasic actions here, and then the equations can be solved with such divisions. In contrast, if we choose (2,3,4) as the nonbasic actions, the equations cannot be solved. It is because if basic action $a_1$ is determined, the equations will be
$$
    f_1(a_2, a_4; a_1)=0, f_2(a_2,a_3)=0, f_3(\emptyset; a_1)=0,
$$
which is unsolvable.

Actually, part of our experiments is executed with random choices on valid divisions, and we do not observe large variances with different valid divisions. We will add the above analyses in the final version.

---

### Decision · Program_Chairs · 2023-09-21

**Decision:**

Accept (poster)

**Comment:**

The authors introduced a policy optimization methodology suitable for continuous control problems with hard constraints. The proposed methodology is innovative by introducing GRG to the RL setting with the goal to create a policy optimization with hard constraints in a problem-agnostic manner to solve a variety of complex tasks. Paper is well-written and easy to read, experiments present good results in comparisons with existing RL baselines.